# Cyprocide selectively kills nematodes via cytochrome P450 bioactivation

Jessica Knox [1,2], Andrew R. Burns [1,2], Brittany Cooke[1,2],
Savina R. Cammalleri[1,2], Megan Kitner[3], Justin Ching [4], Jack M. P. Castelli[1,2],
Emily Puumala[1], Jamie Snider [2], Emily Koury [5], J. B. Collins [5],
Salma Geissah[1,6], James J. Dowling [1,6], Erik C. Andersen[7], Igor Stagljar [1,2,8,9],
Leah E. Cowen [1], Mark Lautens [4], Inga Zasada[3] & Peter J. Roy [1,2,10] ✉

Left unchecked, plant-parasitic nematodes have the potential to devastate crops globally. Highly effective but non-selective nematicides are justifiably being phased-out, leaving farmers with limited options for managing nematode infestation. Here, we report our discovery of a 1,3,4-oxadiazole thioether scaffold called Cyprocide that selectively kills nematodes including diverse species of plant-parasitic nematodes. Cyprocide is bioactivated into a lethal reactive electrophilic metabolite by specific nematode cytochrome P450 enzymes. Cyprocide fails to kill organisms beyond nematodes, suggesting that the targeted lethality of this pro-nematicide derives from P450 substrate selectivity. Our findings demonstrate that Cyprocide is a selective nematicidal scaffold with broad-spectrum activity that holds the potential to help safeguard our global food supply.

The expanding human population, increasing demand for high-quality protein, decreasing arable land, and impacts of climate change constitute significant challenges that jeopardize global food security[1,2]. Compounding these issues, more than 4000 species of plant-parasitic nematodes (PPNs) reduce global crop yield by over 12% annually[3,4]. The phasing out of effective but non-selective chemical nematicides adds additional pressure to our food production pipeline[5-7]. Consequently, there is a pressing need for novel nematicides with improved selectivity.

Our group recently discovered a novel class of imidazothiazole compounds called the Selectivins (e.g., **1**) that selectively kill nematodes via cytochrome P450-mediated bioactivation[8]. Xenobiotic-metabolizing cytochrome P450 enzymes (P450s) typically detoxify their substrates, catalyzing monooxygenation reactions that enhance substrate hydrophilicity and promote metabolite excretion via drug efflux pumps[9]. However, some substrate-P450 pairs transform a

relatively inert compound into a metabolite that exhibits distinct or heightened biological activity[10-12]. This bioactivation can be phylum or species-selective due to the vast phylogenetic diversity of P450s[8,13]. Indeed, we have shown that the Selectivins are selectively bioactivated by specific nematode P450s into reactive electrophilic products that kill nematodes but not non-target species from diverse phyla[8]. Our discovery of the Selectivins established P450-mediated bioactivation as a novel approach to achieve nematode selectivity.

Here, we have re-screened our collection of worm-active molecules for P450-dependent lethality and identified multiple disubstituted oxadiazole (DODA) molecules that we previously found to have nematode-selective activity (e.g., **2**–**4**)[14]. We consequently assembled an expanded library of DODA compounds and screened it against multiple nematodes, including the free-living *Caenorhabditis elegans* and three species of PPNs from distinct genera. This pipeline

[1]Department of Molecular Genetics, University of Toronto, Toronto, ON, Canada. [2]Terrence Donnelly Centre for Cellular and Biomolecular Research, University of Toronto, Toronto, ON, Canada. [3]United States Department of Agriculture – Agricultural Research Service, Horticultural Crops Disease and Pest Management Research Unit, Corvallis, OR, USA. [4]Davenport Research Laboratories, Department of Chemistry, University of Toronto, Toronto, ON, Canada. [5]Molecular Biosciences, Northwestern University, Evanston, IL, USA. [6]Division of Neurology and Program in Genetics and Genome Biology, The Hospital for Sick Children, Toronto, ON, Canada. [7]Department of Biology, Johns Hopkins University, Baltimore, MD, USA. [8]Department of Biochemistry, University of Toronto, Toronto, ON, Canada. [9]Mediterranean Institute for Life Sciences, Meštrovićevo Šetalište 45, HR-21000 Split, Croatia. [10]Department of Pharmacology and Toxicology, University of Toronto, Toronto, ON, Canada. ✉e-mail: peter.roy@utoronto.ca

yielded a broad-acting P450-dependent 1,3,4-oxadiazole thioether with nematode selectivity among species tested that we call cyprocide-B (**5**). We identified the nematode P450s responsible for bioactivating cyprocide-B into an oxidized electrophile. The bioactivated product reacts with low molecular weight (LMW) thiols such as glutathione, implicating LMW thiol depletion as one potential mechanism by which the lethal metabolite kills cells. Importantly, this bioactivation mechanism is observed across diverse nematodes but not in the non-target organisms examined. Thus, the Cyprocide scaffold is a phylum-selective nematicide with the potential to improve food security.

## Results

### Cyprocide selectively kills nematodes

To identify bioactivated compounds within our custom set of uncharacterized nematicidal small molecules[14], we asked whether the small molecule's lethality in the free-living nematode *C. elegans* could be suppressed by disrupting P450 metabolism. To do this, we genetically reduced the activity of the *C. elegans* P450 oxidoreductase (POR) EMB-8, which transfers electrons to all microsomal P450s (see "Methods" for details)[15–19]. We found that 14 of the 87 compounds assayed required POR to exert maximal lethal activity in *C. elegans* (Fig. 1a, Supplementary Data 1). Among these 14 compounds is selectivin-A (**1**), which we have previously shown to be P450-bioactivated[8]. By contrast, the succinate dehydrogenase inhibitor wact-11, which does not require bioactivation for activity[14], was found to kill worms in a POR-independent manner (Fig. 1a). These results suggest that the remaining 13 POR-dependent molecules are metabolized by P450s into a toxic product.

Three of the 13 remaining POR-dependent nematicides (wact-4 (**2**), wact-191 (**3**), and wact-476 (**4**)) have been previously observed to

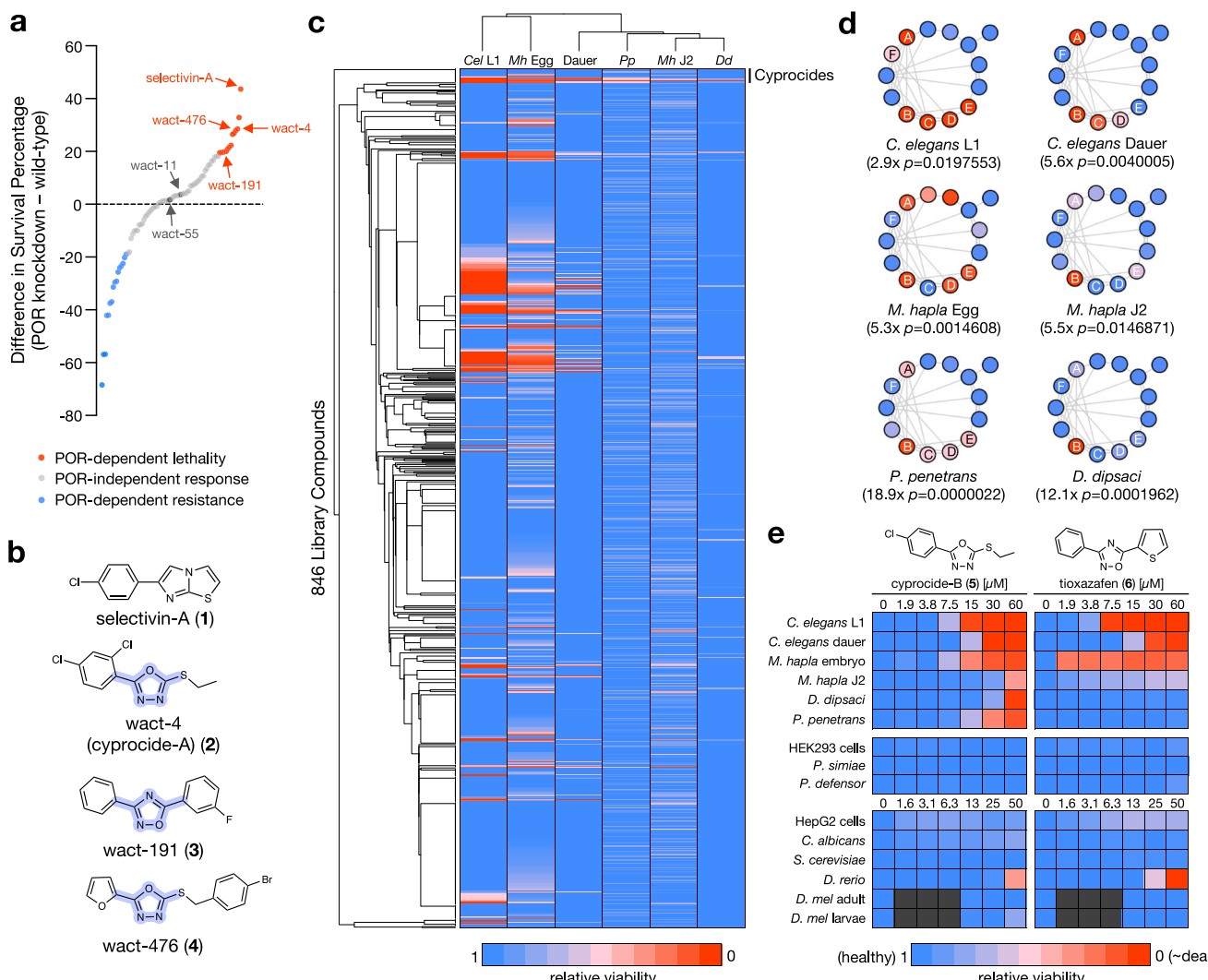

**Fig. 1 | Cyprocide-B is a selective nematicide with broad-spectrum activity in PPNs. a** POR knockdown screen results. Difference in survival percentage between POR knockdown and wild-type conditions is plotted for the 87 unique nematicides tested. **b** Structure of selectivin-A and the three POR-dependent selective nematicides. The shared DODA core scaffold is highlighted in purple. **c** Summarized screening results for the DODA nematicide library in *C. elegans* larval (*Cel* L1) and dauer stages, *M. hapla* embryo (*Mh* Egg) and infective juvenile (*Mh* J2) stages, *D. dipsaci* (*Dd*), and *P. penetrans* (*Pp*). Relative viability is represented by the color-coded scale. Library compounds are organized by structural similarity. **d** The Cyprocides are shown clustered by structural similarity with screening results from (**c**) overlaid. Family members (nodes) are connected by an edge if they share a Tanimoto structural similarity score >0.725. Named analogs are labeled on the nodes. The enrichment of Cyprocides within screen hits and associated one-sided hypergeometric *p*-values are indicated. **e** Dose-response analysis of cyprocide-B and tioxazafen on nematode and non-target organisms, including human cells (HEK293 and HepG2), fungi (*Saccharomyces cerevisiae* and *Candida albicans*), plant beneficial rhizobacteria (*Pseudomonas simiae* and *P. defensor*), zebrafish (*D. rerio*), and *Drosophila melanogaster* (*D. mel*) adult and larval stages. As a positive control for the *D. melanogaster* assays we found 50 μM abamectin killed 100% of adult flies (*n* = 3 biological replicates (BRs)). Compound activity is represented by the color-coded scale. Black indicates the condition was not tested. Source data are provided as a Source data file.

exhibit selective lethality towards nematodes over zebrafish and human HEK293 cells[14] and share a common DODA core scaffold (Fig. 1b). Given the structural similarity and nematicidal properties of these hits, we expanded our exploration of DODA molecules to identify those that might have improved potency and/or broad-spectrum activity against PPNs. We assembled a library of 846 commercially available small molecules with a DODA core that comprised 151 unique scaffolds (Supplementary Data 2). We screened this library at 60 μM for nematicidal activity against *C. elegans* and three economically significant PPN species from distinct genera: the stem and bulb nematode *Ditylenchus dipsaci*, the root lesion nematode *Pratylenchus penetrans*, and the root knot nematode *Meloidogyne hapla* (Fig. 1c, Supplementary Data 2)[20].

These screens revealed that the 2,5-disubstituted-1,3,4-oxadiazole thioether scaffold, which we have named Cyprocide, has multiple analogs that exhibit activity against diverse nematode species (Fig. 1c, d). While wact-4 is a Cyprocide (cyprocide-A) (**2**), we have focused our characterization efforts on cyprocide-B (**5**) (2-(4-chlorophenyl)-5-(ethylthio)-1,3,4-oxadiazole) because it more potently kills PPNs (Fig. 1d, e). Notably, cyprocide-B is relatively inactive against non-nematode phyla at concentrations that kill nematodes in vitro. Non-target organisms including human cells, fungi, plant beneficial rhizobacteria, insects, and fish were tested in dose-response analyses to assess the impact of cyprocide-B on viability alongside the commercial nematicide tioxazafen (**6**) (Fig. 1e). The viability of off-target organisms was relatively unaffected by cyprocide-B except for zebrafish at only the highest tested dose (50 μM). Notably, cyprocide-B was safer for zebrafish than the commercial product tioxazafen (Fig. 1e).

### Cyprocide is bioactivated to a reactive electrophile

Using HPLC[21], we found that cyprocide-B is metabolized into five products (M1-M5, Supplementary Fig. 1A) in wild-type *C. elegans*. In POR-disrupted *C. elegans*, we found significantly less of the cyprocide-B metabolites were produced ($p < 0.01$) (Supplementary Fig. 1A, B) and cyprocide-B's lethality was coincidentally suppressed (Supplementary Fig. 1C). This suppression of lethality via POR disruption extended to other Cyprocide family members but not the wact-11 control[14] (Supplementary Fig. 1C). These findings indicate that cyprocide-B is metabolically converted to a lethal product in *C. elegans*.

LC-MS analysis of cyprocide-B-treated wild-type *C. elegans* lysates revealed that the parent compound likely undergoes an initial P450-catalyzed S-oxidation to generate an electrophilic sulfoxide metabolite (**7**) (Fig. 2a–d). Similar phenyloxadiazole sulfoxides have been shown to react with cysteine thiols in the cell by nucleophilic aromatic substitution, whereas their corresponding thioethers are virtually nonreactive[22]. Soft electrophiles are known to interact with soft nucleophiles such as LMW thiols within cells[23]. Indeed, we found abundant masses in *C. elegans* lysates corresponding to cyprocide-B sulfoxide conjugates with glutathione (**8**), γ-glutamylcysteine (**9**), cysteinylglycine (**10**), and cysteine (**11**) LMW thiols (Fig. 2e–h).

We hypothesized that the lethality of cyprocide-B sulfoxide metabolite is intimately related to its ability to deplete the aforementioned anti-oxidant LMW thiols[11]. To test this hypothesis, we asked whether exogenously supplied N-acetylcysteine ethyl ester (NACET), which is converted by the cell into cysteine, γ-glutamylcysteine, and glutathione[24], can suppress the lethality conferred by cyprocide-B. Indeed, addition of 5 mM NACET to *C. elegans* larvae robustly suppressed cyprocide-B-induced lethality but not that of POR-independent control nematicides wact-11 and wact-55 (Figs. 1a and 2i). These results suggest that the bioactivated cyprocide-B electrophilic metabolite must deplete the LMW thiols to exert its killing effect. LMW thiol depletion likely makes the cell more vulnerable to redox stress[25] and allows the electrophilic metabolite to interfere with essential processes to confer nematicidal activity in vivo.

### Cyprocide is bioactivated by *C. elegans* CYP-35D1

To identify the P450 responsible for cyprocide-B bioactivation in *C. elegans*, we asked whether the RNAi-knockdown of any of 69 of the 76 *C. elegans* P450 (cyp) genes[26] suppressed cyprocide-B lethality. We found that RNAi-knockdown of *cyp-35D1* conferred significant resistance to cyprocide-B ($p < 0.0001$) (Fig. 3a). This result was confirmed using two distinct deletion mutants of *cyp-35D1* (Fig. 3b, Supplementary Fig. 2). Hence, CYP-35D1 is necessary for cyprocide-B bioactivity.

We next asked whether CYP-35D1 is sufficient for metabolizing cyprocide-B into a lethal metabolite. We expressed CYP-35D1 in *Saccharomyces cerevisiae* yeast and tested its ability to metabolize cyprocide-B and coincidentally induce lethality. Indeed, heterologous CYP-35D1 expression hypersensitized yeast to cyprocide-B (Fig. 3c) and induced the production of the LMW thiol conjugates of cyprocide-B (Fig. 3d). Together, these results demonstrate that CYP-35D1 is both necessary and sufficient for cyprocide-B metabolism into a lethal product.

### Cyprocide is bioactivated by P450s in diverse PPN species

We profiled cyprocide-B metabolite production in lysates of PPN species *Ditylenchus dipsaci, Pratylenchus penetrans*, and *Meloidogyne hapla* after exposure to 100 μM cyprocide-B using LC-MS. Cyprocide-B-LMW thiol conjugates were produced in all PPNs tested, indicating a P450-catalyzed metabolic bioactivation had occurred similar to what was observed in *C. elegans* (Supplementary Fig. 3). The reactive sulfoxide metabolite of cyprocide-B was either not detected or observed in low abundance, suggesting that it was efficiently consumed by the LMW thiols in the PPNs, which is consistent with the efficient conjugation of xenobiotic electrophiles seen in other species[8,24].

Four lines of evidence indicate that Cyprocide's lethality in PPNs is P450-dependent. First, we tested whether the small molecule pan-P450 inhibitor 1-aminobenzotriazole (1-ABT)[27] can suppress cyprocide-B-induced lethality in *C. elegans* and four PPN species and found significant suppression in all species tested ($p < 0.0001$) (Supplementary Fig. 4). Second, we examined the impact of 1-ABT on metabolite production in the PPN *D. dipsaci*. We found 1-ABT exposure caused an increased abundance of unmodified cyprocide-B parent and a corresponding reduction in metabolite abundance ($p < 0.05$) (Fig. 4a–d). Third, we found that preincubation of *D. dipsaci* with NACET significantly suppressed the lethality induced by cyprocide-B ($p < 1e-6$) but not that conferred by the fluopyram nematicide control (Fig. 4e). Here, fluopyram was selected as a negative control because fluopyram kills *D. dipsaci* and inhibits its target without the need for bioactivation[14]. Finally, we sought to identify a P450 in PPNs that can carry out Cyprocide bioactivation using the notorious root knot nematode *Meloidogyne incognita* as a representative species. To do this experiment, we heterologously expressed 19 representative *M. incognita* P450s in yeast and measured the impact of cyprocide-B on strain growth (Fig. 4f). Yeast expressing CYP4731A3 grew significantly slower in response to cyprocide-B relative to the control ($p < 0.001$) (Fig. 4f, g). LC-MS analysis revealed that *M. incognita* CYP4731A3-expressing yeast robustly metabolized cyprocide-B, producing the same LMW thiol conjugates that were found in the CYP-35D1-expressing yeast strain (Fig. 4h, i). Together, these results indicate that P450-mediated bioactivation of Cyprocide is necessary and sufficient to generate reactive lethal metabolites in diverse impactful PPN species.

### Cyprocide controls root infestation by *M. incognita*

To further explore Cyprocide's utility against PPNs, we asked whether the nematicide can maintain its activity in soil where many candidate nematicides lose efficacy because of rapid degradation or poor mobility[28]. We pre-drenched 90 grams of soil with a 60 μM aqueous solution of small molecule to assay the

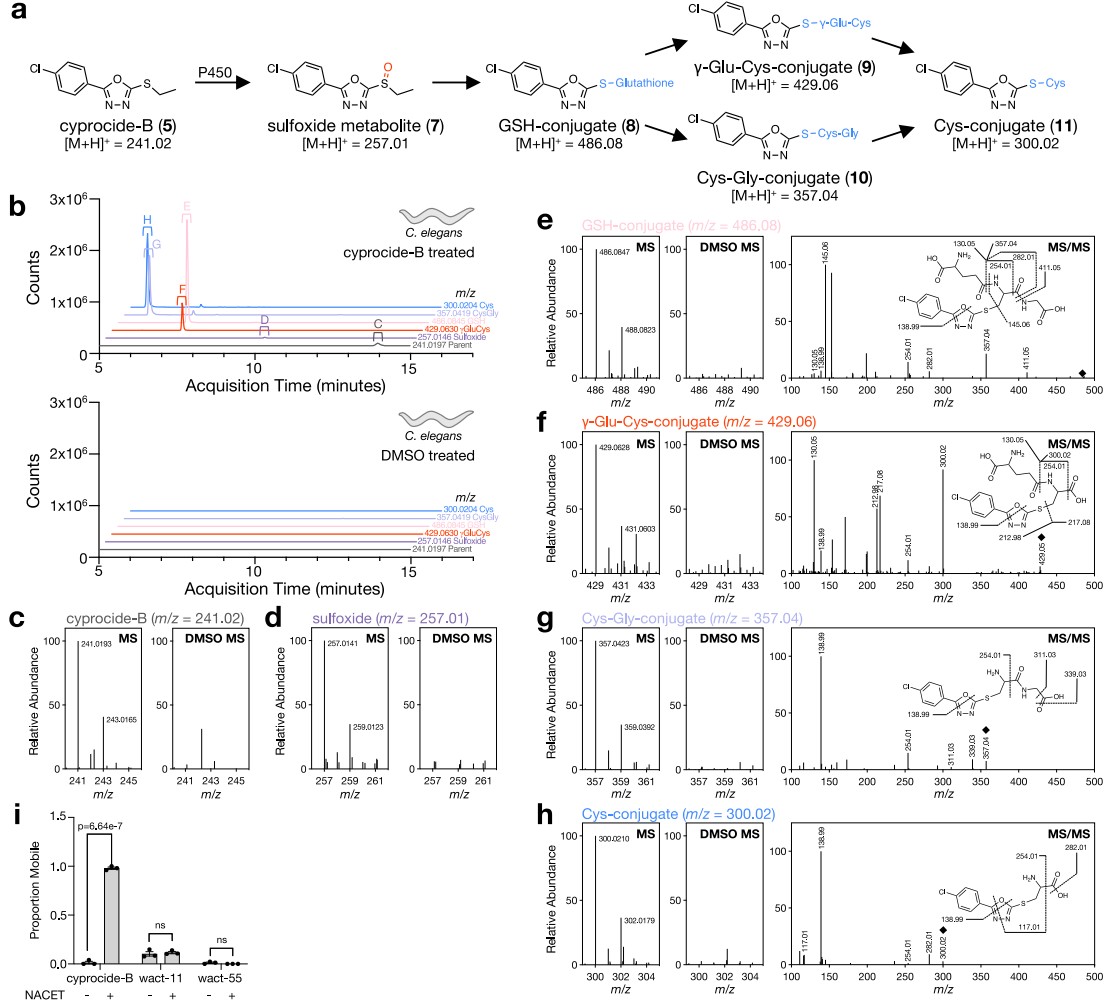

**Fig. 2 | Cyprocide-B is bioactivated to produce a reactive electrophile.**
**a** Schematic of the proposed cyprocide-B sulfoxidation and LMW thiol conjugation pathway. P450s likely catalyze the S-oxidation of cyprocide-B, which produces an electrophilic sulfoxide metabolite. This sulfoxide metabolite reacts with glutathione (GSH) and is likely further processed to γ-glutamylcysteine (γ-Glu-Cys), cysteinylglycine (Cys-Gly) and cysteine (Cys) conjugates. The cyprocide-B sulfoxide metabolite can also react directly with γ-glutamylcysteine, cysteinylglycine and/or cysteine. **b** Extracted ion chromatograms (EICs) for indicated m/z values found in cyprocide-B treated *C. elegans* lysate (above) and paired DMSO solvent control (below). **c–h** Mass spectrometry (MS) data for the unmodified cyprocide-B parent

(**c**), the sulfoxide metabolite (**d**), and the GSH-, γ-Glu-Cys-, Cys-Gly- and Cys- conjugates (**e–h**) from lysates of *C. elegans* treated with cyprocide-B and paired solvent-treated controls. The predicted structures of the thiol conjugates are supported by tandem mass spectrometry (MS/MS) (**e–h**). **i** *C. elegans* L1 larvae were preincubated with NACET or solvent control for 4 h before a 24-h exposure to 25 μM cyprocide-B, wact-11, or wact-55. The proportion of mobile worms in each condition is reported ($n = 3$ BRs). Error bars indicate SEM, $p$-values obtained from unpaired two-tailed Student's t-tests, comparing means of the NACET and solvent preincubation conditions, ns is not significant ($p = 0.6352$ for wact-11 and $p = 0.1496$ for wact-55). Source data are provided as a Source data file.

ability of the Cyprocides to protect a tomato host plant from 500 *M. incognita* infective juveniles in a standard root infection assay[29]. The Cyprocides were tested alongside two next-generation soil-applied commercial nematicide controls, fluopyram and tioxazafen, which have proven to be effective at controlling *M. incognita* infestation in similar assays[8]. After 6 weeks of infection, the number of eggs per milligram of root tissue was measured. We found that cyprocide-E reduced root egg burden by an average of 73% relative to untreated controls ($p < 0.05$) (Fig. 4j), outperforming tioxazafen (53% reduction in eggs/mg root tissue) but not fluopyram (100% reduction in eggs/ mg root tissue). The presence of LMW thiol conjugates of cyprocide-E in *M. incognita* CYP4731A3-expressing yeast lysates suggests that cyprocide-E is bioactivated similarly to cyprocide-B (Supplementary Fig. 5). The Cyprocides did not negatively impact tomato root growth relative to solvent controls ($p > 0.05$) (Fig. 4k). Hence, the Cyprocides have potential utility in preventing root infestation by PPNs in a soil environment.

## A structure-activity analysis reveals cyprocide analogs with increased potency
Our DODA screen revealed six Cyprocide analogs (Fig. 1d, cyprocide-A-F) that yielded a serendipitous set of structure-activity relationships (SAR). In the hopes of identifying molecules with increased potency, the SAR was expanded to an additional 23 rationalized analogs that vary in structure on either side of the 1,3,4-oxadiazole thioether core (Supplementary Fig. 6). The molecules were either procured from commercial sources or synthesized in-house and screened against three nematode species (Supplementary Fig. 6). Because Cyprocide's mode-of-action necessitates testing analogs in living systems, differences in analog activity in the different species likely reflect not only P450 engagement but also absorption, distribution, metabolism (by other enzymes), and drug-export (ADME). We found that the trifluoromethylated cyprocide-I analog exhibited eight-fold greater potency in vitro against *M. hapla* compared to cyprocide-B. Cyprocide-I's increased potency was consistent with Cyprocide's

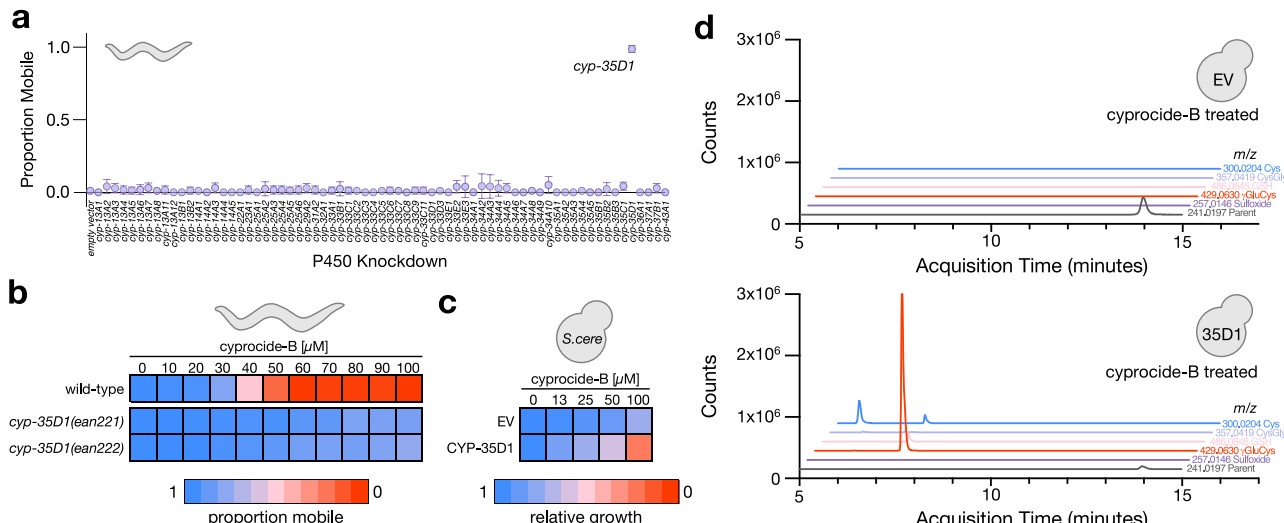

**Fig. 3 | Cyprocide-B is bioactivated by *C. elegans* CYP-35D1. a** P450 RNAi knockdown screen results. The proportion of mobile adult *C. elegans* worms is reported in each knockdown condition after a 48-h exposure to 40 μM cyprocide-B (*n* = 4 technical replicates (TRs)). Error bars indicate standard deviation, *p*-value comparing the mean of the empty vector to *cyp-35D1* knockdown condition (*p* = 1.30e−9) was obtained from an unpaired two-tailed Student's t-test. **b** Cyprocide-B dose-response in adult *C. elegans*. The proportion of mobile worms in each condition is indicated by the color-coded scale in the wild-type and *cyp-35D1* deletion mutants after a 4-day exposure to the indicated concentrations of cyprocide-B (*n* = 3 BRs). **c** Cyprocide-B dose-response for a *S. cerevisiae* strain expressing *C. elegans* CYP-35D1 and one carrying an empty vector (EV) that does not express a P450. The mean area under the growth curve in each condition over 48-h of cyprocide-B exposure relative to solvent controls is indicated by the color-coded scale (*n* = 4 BRs). **d** Yeast expressing CYP-35D1 and the EV control strain were exposed to 100 μM cyprocide-B for 6 h. Yeast lysates were analyzed by LC-MS and extracted ion chromatograms for the indicated cyprocide-B parent and metabolite *m/z* values are shown for both EV and CYP-35D1-expressing strains. Source data are provided as a Source data file.

mode-of-action whereby sulfoxidation increased the electrophilicity of the oxadiazole C5. The trifluoromethyl leaving group's increased electron withdrawing capability should further increase the electrophilicity of the oxadiazole's C5 upon P450-mediated sulfoxidation, enhancing reactivity of the metabolite. Molecules with the trifluoromethylated R² substituent maintained high potency when combined with distinct R¹ electron withdrawing groups (cyprocide-N and -N-2, cyprocide-U and -U-2), suggesting that many potent cyprocide analogs can be generated with different physicochemical properties and perhaps biological ones as well.

## Discussion

Here we report our discovery and characterization of the Cyprocide scaffold of selective nematicides. The otherwise inert Cyprocides are bioactivated by P450s across three genera of PPN species yet have limited activity in the non-target organisms tested. This suggests that nematode-selectivity among the species tested is derived from the potentially unique ability of nematode P450s to metabolize the Cyprocides into electrophilic products. This mode-of-action is shared with the recently discovered Selectivin class of nematicides[8] and provides further evidence that P450-catalyzed bioactivation is a viable mechanism through which nematode-selectivity can be achieved with distinct chemical structures.

Members of both the Cyprocide and Selectivin classes of pro-nematicide are active in vitro and in soil-based tomato infection assays against PPNs from the *Meloidogyne* genus. We have found the Cyprocides to have a broader spectrum of in vitro nematicidal activity, affecting both *Pratylenchus* and *Ditylenchus* nematodes in a P450-dependent manner. Thus, the Cyprocide scaffold may hold potential utility to control a vast diversity of parasitic nematodes, but it remains to be seen if this broad-spectrum activity extends to soil-based infestation assays. Despite the activity against all nematodes tested, the Cyprocides are relatively inactive against non-target organisms such as plant-growth-promoting rhizobacteria that are increasingly being

incorporated into integrated pest management strategies alongside pesticide controls[30].

The Cyprocides exhibit similar or improved selectivity for nematodes over many of the commercial nematicides previously tested in these non-target organism assays[8]. However, as evidenced throughout this study, many of the Cyprocides examined are lethal to the free-living nematode *C. elegans* and thus may potentially pose a threat to soil-beneficial nematode communities. While the same can be said of other next-generation commercial nematicides including tioxazafen and fluopyram[8,14], future efforts can be taken to identify Cyprocide analogs that minimize detrimental effects on free-living nematodes while maintaining activity in PPNs.

Both Cyprocide and Selectivin[8] are bioactivated by *M. incognita*'s CYP4731A3. This P450 has high basal expression in infective juveniles[31], making it an ideal bioactivating enzyme to target for the identification of phylum-selective pro-nematicides. Given the structural distinctiveness of Cyprocide and Selectivin, CYP4731A3 is likely capable of metabolizing additional structurally diverse substrates. Yeast heterologously expressing CYP4731A3 are a convenient system with which to identify such molecules, which is especially important in our search for control agents against pests that are difficult to culture in the laboratory.

The difference in agricultural output between fields infested with PPNs and those managed with nematicides is stark[3,4]. Given the phase-out of non-selective nematicides[5–7] and the potential for the development of resistance to some of the next-generation nematicides[32,33], additional selective nematicides are needed. The discovery of the Cyprocides has reinforced the idea that P450 bioactivation is a viable mechanism through which this need may be met. Further development of the scaffold may ultimately help to secure global food production.

## Methods

The research reported herein complies with all relevant ethical regulations. All zebrafish experiments were performed in compliance with any relevant ethical regulations, specifically following an institutionally

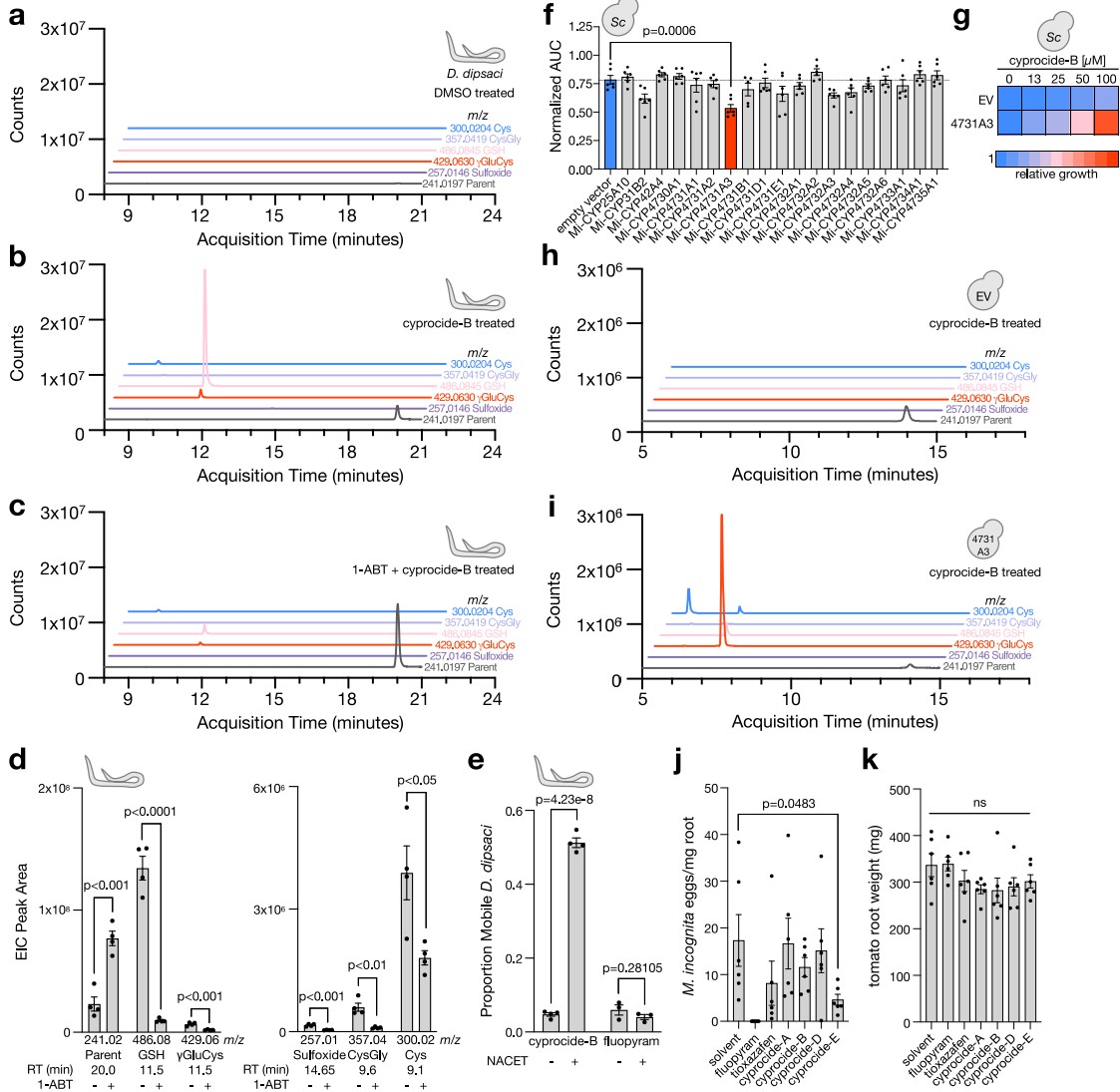

**Fig. 4 | Cyprocide-B is bioactivated into a toxic electrophile by P450s in PPNs.** EICs for indicated cyprocide-B parent and metabolite *m/z* values from *D. dipsaci* buffer after exposure to **a** DMSO, **b** cyprocide-B, and **c** 1-ABT plus cyprocide-B. **d** EIC peak area at indicated retention time (RT) for cyprocide-B parent and metabolites in *D. dipsaci* buffer after incubation with cyprocide-B+/− 1-ABT (*n* = 4 BRs). Exact *p*-values are as follows: Parent (*p* = 0.000642), GSH (*p* = 0.000014), γGluCys (*p* = 0.000757), Sulfoxide (*p* = 0.000236), CysGly (*p* = 0.002462), Cys (*p* = 0.022239). **e** *D. dipsaci* were preincubated +/− NACET before a three-day incubation in cyprocide-B or fluopyram. The proportion of mobile worms after chemical exposure is reported (*n* = 4 BRs for cyprocide-B exposures, *n* = 3 BRs for fluopyram exposures). **f** Mean AUC of *M. incognita* P450-expressing yeast strains exposed to 50 µM cyprocide-B, normalized to the solvent control for each strain (*n* = 6 BRs). **g** Cyprocide-B dose-response for yeast strains expressing CYP4731A3 and EV controls. Mean AUC relative to solvent controls is indicated by the color-coded scale (*n* = 4 BRs). **h, i** EICs for indicated cyprocide-B parent and metabolite *m/z* values from yeast lysates after cyprocide-B exposure in EV and CYP4731A3-expressing strains. **j** Number of *M. incognita* eggs per milligram of tomato root tissue after 6 weeks of infection in soil drenched with solvent or indicated chemical at 60 µM is reported (*n* = 6 BRs). **k** Tomato root weight at infection assay endpoint in each condition (*n* = 6 BRs). In all panels error bars represent SEM, *p*-values were obtained from unpaired two-tailed Student's t-tests, ns is not significant (*p* > 0.05). Source data are provided as a Source data file.

reviewed and approved animal use protocol as well as the policies and guidelines of the Canadian Council on Animal Care and Animals for Research Act of Ontario.

### *Caenorhabditis elegans* strains and culture methods

The *Caenorhabditis elegans* wild-type (N2), CB1370 *daf-2(e1370)* and MJ69 *emb-8(hc69)* strains were obtained from the *Caenorhabditis* Genetics Center (CGC, University of Minnesota). The *cyp-35D1(ean221)* and *cyp-35D1(ean222)* deletion mutants were generated in the lab of Erik Andersen (Johns Hopkins University). *C. elegans* N2 and the *cyp-35D1* mutants were cultured on MYOB media using standard methods at 20 °C[34]. The temperature sensitive *daf-2(e1370)* and *emb-8(hc69)*

mutants were cultured on MYOB at the permissive temperature of 16 °C.

### Commercial chemical sources

The POR-dependency screen compounds were purchased from ChemBridge Corporation (Supplementary Data 1 for details). The DODA library compounds were purchased from ChemBridge Corporation and MolPort (Supplementary Data 2 for details). Cyprocide-A,-C,-D,-E, and -F analogs and tioxazafen were purchased from ChemBridge Corporation. Cyprocide-B was purchased from ChemBridge Corporation and Vitas-M. Cyprocide-H, -M, -N, -Q, -W, -Y, and -M-2 were purchased from MolPort. Fluopyram and 1-Aminobenzotriazole (1-

ABT) were purchased from MilliporeSigma. N-acetyl-L-Cysteine ethyl ester (NACET) was purchased from Cayman Chemical.

## *C. elegans* POR-dependency screen

For the wild-type chemical screens, synchronized first larval stage (L1) N2 worms were plated on nematode growth media (NGM) agar plates[35] supplemented with 1 mM IPTG and 100 μg/mL carbenicillin and seeded with HT115(DE3) *Escherichia coli* carrying the empty L4440 RNAi feeding control vector. The bacteria used to seed the NGM plates were grown overnight in LB media supplemented with 100 μg/mL ampicillin at 25 °C in 50 mL conical tubes, without aeration. Once the cultures achieved an $OD_{600}$ of 0.6, 1 mM of IPTG was added and the cultures were incubated on a nutating shaker at 20 °C for 1 h. After IPTG induction, the cultures were concentrated five-fold with double-distilled water (ddH$_2$O), and then seeded onto the solid NGM plates. The following day the plates were dried for 45 min in a laminar flow hood. After drying, synchronized L1s that were obtained from an alkaline bleach treatment of gravid adults (embryo preparation)[36] were deposited onto the plates. The worms were grown for 48 h at 25 °C until they reached the late L4 stage. The L4 worms were then washed off of the plates with M9 buffer[35], and worm density was adjusted to 2.5 worms per microliter. Screening plates were prepared by adding 40 μL of HT115(DE3) + L4440 bacteria suspended in liquid NGM to each well of several 96-well flat-bottom culture plates, pinning 0.3 μL of each library chemical (at 5 mM) into the corresponding wells, and then adding 10 μL of the worm suspension (~25 worms) to each well. The final chemical concentration in the wells was 30 μM. The bacteria for the liquid-based screening plates were grown and prepared the same way as for the solid NGM plates (see above), except after IPTG induction the cultures were concentrated with liquid NGM media instead of water. The screening plates were sealed with parafilm and stored in a box containing several paper towels soaked with water. The plates were incubated for one day at 25 °C with shaking at 200 rpm. After incubation, worm viability was assessed. Relatively clear worms, with poorly defined internal structures, that failed to move after vigorous jostling of the plate were scored as dead. The percent of living worms in each well was calculated. The *emb-8* knockdown screen was performed identically to the wild-type screen, except the MJ69 strain was used in place of the wild-type strain, and worms were fed HT115(DE3) *E. coli* harboring an RNAi feeding vector expressing dsRNA targeting the *emb-8* gene from the Ahringer *C. elegans* RNAi feeding library distributed by Source BioScience Ltd.

As our POR knockdown protocol necessitated screening in young adult worms, we selected a subset of our uncharacterized nematicide collection[14] that killed ≥20% of young adult *C. elegans* at 30 mM to include in the POR-dependency screen. This step left us with a focused library of 87 compounds, comprising 67 distinct structural scaffolds which we assayed in both wild-type and POR knockdown conditions to assess the impact of P450-catalyzed metabolism on nematicidal activity in the worm. From 68 untreated control wells we found that the average survival rate for EMB-8 knockdown worms is 98.6%. Since our focused library of nematicides was constructed such that at most 80% of nematicide-treated worms will survive, we reasoned that an increase in survival of 18.6 percentage points or more for *emb-8* knockdown worms, relative to wild-type, is indicative of EMB-8-dependent killing. These nematicides require functional EMB-8 for activity and are putatively bioactivated by P450 enzymes in the worms. Conversely, a decrease in survival by 18.6 percentage points or more is indicative of EMB-8-dependent resistance, i.e., worms require EMB-8 to resist the nematicidal activity of the compounds. These nematicides are putatively detoxified by worm P450s.

## DODA library small molecule screening methods

### *C. elegans* L1 survival and reproduction screen.
The DODA library screening method was adapted from previously described liquid-based screening protocols[14]. *E. coli* strain HB101 grown overnight in LB was resuspended in an equal volume of liquid NGM. 40 μL of bacterial suspension was dispensed into each well of a flat-bottomed 96-well culture plate, and 0.3 μL of the small molecules or dimethyl sulfoxide (DMSO) solvent control was pinned into the wells using a 96-well pinning tool (V&P Scientific). Approximately 20 synchronized first-stage larvae (L1) wild-type N2 worms obtained from an embryo preparation on the previous day were then added to each well in 10 μL M9. The final concentration of library compound in the wells was 60 μM (0.6% v/v DMSO). Plates were sealed with Parafilm and incubated for 6 days at 20 °C while shaking at 200 rpm (New Brunswick I26/I26R shaker, Eppendorf). On day six the plates were observed under a dissection microscope and the survival and reproduction phenotypes were assessed in each condition. In solvent exposure conditions we observe that by day 6 the deposited parental larvae have reached adulthood and produced numerous (≫50) next-generation larvae in the well. Two biological replicates of the screen were performed in a single technical replicate. A scoring system was used to summarize these phenotypes where a score of 0 indicates 0–10 worms were living in the well, 1 indicates 11–20 worms, 2 indicates 21–50 worms, 3 is greater than 50 worms and 4 indicates the well is overgrown and starved of bacterial food with many more than 50 worms in the well. Hits were defined as any compound for which the average score of the two replicates was 1 or lower.

### *C. elegans* dauer mobility screen.
*C. elegans* strain CB1370 *daf-2(e1370)* synchronized L1s were obtained from an embryo preparation that was left to hatch at 15 °C overnight. The L1s were plated on 10 cm MYOB media plates (5000 L1s per plate) seeded with *E. coli* OP50 and transferred to 25 °C for 48 h to progress into dauer stage (100% dauer formation is expected at 25 °C). 40 μL of NGM was dispensed into each well of a 96-well plate, and 300 nL of the small molecules or DMSO solvent control was pinned into the wells using a 96-well pinning tool (V&P Scientific). Dauer larvae were washed off of the MYOB plates using M9 and approximately 20 dauer larvae were added to each well of the 96-well plates in 10 μL M9. The final concentration of library compound in the wells was 60 μM (0.6% v/v DMSO). Plates were sealed with Parafilm and incubated for 5 days at 20 °C while shaking at 200 rpm. On day five the plates were observed under a dissection microscope and proportion of mobile (living) worms was quantified in each condition. Two biological replicates of the screen were performed in a single technical replicate and the mobility scores for these two replicates were averaged. The mean proportion mobile (relative to DMSO controls) and standard deviation amongst all library compounds was calculated and used to generate Z-scores for each compound screened. Hits were defined as any compound with a Z-score of −2 of lower, which correlated to a proportion mobile of 0.63 or lower.

### *Ditylenchus dipsaci* mobility screen.
The *D. dipsaci* strain G-137 used in this work was collected from garlic in Prince Edward County, Ontario, Canada and was provided to us by Qing Yu (Agriculture and Agri-Food Canada). *D. dipsaci* was cultured and collected for screening as previously described[37]. Briefly, *D. dipsaci* was maintained on pea plants in Gamborg B-5 agar media. Six to eight weeks after inoculation, the *D. dipsaci* were extracted from the plates for use in small molecule screens. The agar plate and pea plant tissue were cut into small cubes and placed in a coffee filter lined funnel. The funnel was placed in a beaker filled with distilled water, allowing the worms to travel through the filter and into the collection beaker. The next day, the *D. dipsaci* worms in the collection beaker were quantified and were ready for use in the chemical screen. To prepare the screen plates, 40 μL of distilled water was dispensed into each well of a 96-well plate, and 0.3 μL of the small molecules or DMSO solvent control was pinned into the wells using a 96-well pinning tool (V&P Scientific). Approximately 20 *D. dipsaci* worms were added to each well of the 96-well plates in 10 μL

distilled water. The final concentration of library compound in the wells was 60 µM (0.6% v/v DMSO). Plates were sealed with Parafilm and incubated for 5 days at 20 °C while shaking at 200 rpm. On day five, the plates were observed under a dissection microscope and the proportion of mobile (living) worms was quantified in each condition after adding 2 µL of 1 M NaOH to each well to stimulate movement. Two biological replicates of the screen were performed in a single technical replicate and the mobility scores for these two replicates were averaged. The mean proportion mobile (relative to DMSO controls) and standard deviation amongst all library compounds was calculated and used to generate Z-scores for each compound screened. Hits were defined as any compound with a Z-score of −2 of lower, which correlated to a proportion mobile of 0.86 or lower.

**Meloidogyne hapla egg hatching screen.** The *M. hapla* strain used in this work was collected from a muck soil sample in Ste-Clotilde, Quebec, Canada and was provided to us by Benjamin Mimee (Agriculture and Agri-Food Canada). *M. hapla* was maintained on tomato (*Solanum lycopersicum* -Rutgers) plants and eggs were collected from tomato roots as previously described[38]. To prepare the screen plates, 40 µL of distilled water was dispensed into each well of a 96-well plate, and 0.3 µL of the small molecules or DMSO solvent control was pinned into the wells using a 96-well pinning tool (V&P Scientific). Approximately 75 *M. hapla* eggs were added to each well of the 96-well plates in 10 µL distilled water. The final concentration of library compound in the wells was 60 µM (0.6% v/v DMSO). Plates were sealed with Parafilm and incubated for 12 days at 25 °C. On day 12, the plates were observed under a dissection microscope and the number of hatched J2s was quantified in each condition. The relative egg hatching score for each library compound was calculated by dividing the number of eggs hatched in the compound well by the average of the number of eggs hatched in the eight DMSO solvent control wells on the same 96-well plate. Two biological replicates of the screen were performed in a single technical replicate and the egg hatching scores for these two replicates were averaged. The mean egg hatching score and standard deviation amongst all library compounds was calculated to generate Z-scores for each compound screened. Hits were defined as any compound with a Z-score of −2 of lower, which correlated to an egg hatching score relative to DMSO of 0.21 or lower.

**Meloidogyne hapla J2 mobility screen.** Infective second-stage juveniles (J2) of *M. hapla* were collected as previously described[38]. To prepare the screen plates, 40 µL of distilled water was dispensed into each well of a 96-well plate, and 0.3 µL of the small molecules or DMSO solvent control was pinned into the wells using a 96-well pinning tool (V&P Scientific). Approximately 25 *M. hapla* J2s were added to each well of the 96-well plates in 10 µL distilled water. The final concentration of library compound in the wells was 60 µM (0.6% v/v DMSO). Plates were sealed with Parafilm and incubated for 5 days at 20 °C while shaking at 200 rpm. On day five, the plates were observed under a dissection microscope and the proportion of mobile (living) worms was quantified in each condition after adding 2 µL of 1 M NaOH to each well to stimulate movement. The relative proportion mobile for each library compound was calculated by dividing the mobility score in the compound exposure by the average of the mobility scores in the eight DMSO control wells on the same 96-well plate. Two biological replicates of the screen were performed in a single technical replicate and the mobility scores for these two replicates were averaged. The mean relative proportion mobile and standard deviation amongst all library compounds was calculated and used to generate Z-scores for each compound screened. Hits were defined as any compound with a Z-score of −2 of lower, which correlated to a proportion mobile of 0.70 or lower.

**Pratylenchus penetrans mobility screen.** The *P. penetrans* strain used in this work was provided to us by Benjamin Mimee (Agriculture and

Agri-Food Canada) and was cultured and collected for screening as previously described for *D. dipsaci*[37] with modifications: *P. penetrans* cultures were maintained on excised Golden Jubilee corn roots (OSC Seeds) in Gamborg B-5 agar media (8 g/L agar). Six to eight weeks after inoculation, the *P. penetrans* were extracted from the plates for use in small molecule screens. The agar plate and corn root tissue were cut into small cubes and placed in a coffee filter lined funnel. The funnel was placed in a beaker filled with distilled water, allowing the worms to travel through the filter and into the collection beaker. The next day, the *P. penetrans* worms in the collection beaker were quantified and were ready for use in the chemical screen. To prepare the screen plates, 40 µL of distilled water was dispensed into each well of a 96-well plate, and 0.3 µL of the small molecules or DMSO solvent control was pinned into the wells using a 96-well pinning tool (V&P Scientific). Approximately 20 *P. penetrans* worms were added to each well of the 96-well plates in 10 µL distilled water. The final concentration of library compound in the wells was 60 µM (0.6% v/v DMSO). Plates were sealed with Parafilm and incubated for 5 days at 20 °C while shaking at 200 rpm. On day five, the plates were observed under a dissection microscope and proportion of mobile (living) worms was quantified in each condition. The relative proportion mobile for each library compound was calculated by dividing the mobility score in the compound exposure by the average of the mobility scores in the eight DMSO control wells on the same 96-well plate. Two biological replicates of the screen were performed in a single technical replicate and the mobility scores for these two replicates were averaged. The mean relative proportion mobile and standard deviation amongst all library compounds was calculated and used to generate Z-scores for each compound screened. Hits were defined as any compound with a Z-score of −2 of lower, which correlated to a proportion mobile of 0.74 or lower.

## Cheminformatic analysis

Molecule scaffolds were generated from chemical SMILES provided by vendors for all DODA library molecules using the Strip-it command line tool version 1.0.2 (RINGS_WITH_LINKERS_1 scaffold definition). The ChemmineR Cheminformatics Toolkit for R package[39] was used to create an atom pair distance matrix of these scaffolds and perform hierarchical clustering to create a dendrogram display of the library organized by structural similarity. The full chemical structures of the Cyprocide scaffold molecules in the DODA library were analyzed for pairwise structural similarity. Similarity scores were calculated as the Tanimoto coefficient of shared FP2 fingerprints using OpenBabel (http://openbabel.org). The visualization of scaffold similarity in Fig. 1d was generated using Cytoscape version 3.9.1[40] with individual Cyprocide compounds (nodes) connected by an edge if they share a structural similarity (Tanimoto coefficient) of >0.725.

## Dose-response experiments

**C. elegans L1 dose-response assays.** The dose-response experiments on *C. elegans* L1s were set up using the same methodology as described for the DODA library screen with the exception that the cyprocide-B and tioxazafen assays displayed in Fig. 1e the proportion of viable worms in each condition was quantified after 3 days of chemical exposure. The relative proportion of viable worms in each chemical exposure condition was calculated by dividing the proportion viable worms in that condition to the proportion mobile worms in the DMSO controls. At least two biological replicates were performed with three technical replicates on each day and the relative proportion viable worms was averaged across the replicates.

**C. elegans dauer dose-response assays.** The dose-response experiments on *C. elegans* dauers were set up using the same methodology as described for the DODA library screen with the exception that the dose-response assays were quantified on day three of chemical

exposure after adding 1 μL of 1 M NaOH to each well to stimulate movement. The relative proportion mobile (living) worms in each chemical exposure condition was calculated by diving the proportion mobile worms in that condition to the proportion mobile worms in the DMSO controls. Three replicates were performed, and the relative proportion mobile was averaged across the replicates.

**D. dipsaci dose-response assays.** The dose-response experiments on D. dipsaci were set up using the same methodology as described for the DODA library screen with the exception that for the cyprocide-B and tioxazafen assays displayed in Fig. 1e the proportion of mobile (living) worms in each condition was quantified after 3 days of chemical exposure. The relative viability for each compound exposure condition was calculated by dividing the proportion mobile in that condition by the proportion mobile worms in the DMSO solvent controls. At least two biological replicates were performed with three technical replicates on each day and the relative proportion mobile was averaged across the replicates.

**M. hapla egg dose-response assays.** The dose-response experiments on M. hapla eggs were set up using the same methodology as described for the DODA library screen with the following modifications: the final volume of distilled water in each well of the 96-well plate was 100 μL and the compounds were added to the wells using a multichannel pipette to a final DMSO concentration of 0.6% (v/v). The dose-response assays were quantified after 12 days of chemical exposure and the data is presented as the number of eggs that hatched in each condition relative to the number that hatched in DMSO controls. Three biological replicates were performed with at least two technical replicates on each day and the relative egg hatching was averaged across the replicates.

**M. hapla J2 dose-response assays.** The dose-response experiments on M. hapla J2s were set up using the same methodology as described for the DODA library screen with the exception that for the cyprocide-B and tioxazafen assays displayed in Fig. 1e the proportion of mobile (living) worms in each condition was quantified after 3 days of chemical exposure. The relative viability for each compound exposure condition was calculated by dividing the proportion mobile in that condition by the proportion mobile worms in the DMSO solvent controls. At least two biological replicates were performed with three technical replicates on each day and the relative proportion mobile was averaged across the replicates.

**P. penetrans dose-response assays.** The dose-response experiments on P. penetrans were set up using the same methodology as described for the DODA library screen with the exception that the dose-response assays were quantified after 3 days of chemical exposure. The relative viability for each compound exposure condition was calculated by dividing the proportion mobile in that condition by the proportion mobile worms in the DMSO solvent controls. Three biological replicates were performed with at least three technical replicates on each day and the relative proportion mobile was averaged across the replicates.

**Wild-type Saccharomyces cerevisiae dose-response assays.** An overnight saturated culture of S. cerevisiae BY4741 was diluted in YPD to an $OD_{600}$ of 0.015 and 100 μL of culture was dispensed into each well of 96-well flat-bottom culture plates. Plates were incubated for 4 h at 30 °C with shaking at 140 rpm. Compounds were added to the wells using a multichannel pipette to a final DMSO concentration of 1% (v/v). The plates were sealed with clear plastic adhesive film and incubated at 30 °C with intermittent shaking in a microplate spectrophotometer (TECAN) for 24 h. Endpoint growth was quantified by measuring optical density (OD) at 600 nm. The relative growth in each chemical

exposure condition was calculated by dividing the endpoint $OD_{600}$ reading in by the endpoint $OD_{600}$ reading in the DMSO solvent control. Three biological replicates were performed, and the relative growth was averaged across the replicates.

**Candida albicans dose-response assays.** Compound potency against C. albicans (SN95) was assessed using two-fold dose-response assays following a standard protocol. Briefly, YPD medium was inoculated with ~1 × 10³ cells/mL from saturated overnight cultures. Assays were performed in 384-well, flat-bottom microtiter plates (Corning) in a final volume of 40 μL/well. Each compound was added to wells in a two-fold concentration gradient from 50 μM to 0 μM. Plates were then incubated in the dark at 30 °C under static conditions for 48 h. Endpoint growth was quantified by measuring $OD_{600}$ using a spectrophotometer (Molecular Devices) and the results were corrected for background media. Relative fungal growth for each compound treatment was defined by normalizing the $OD_{600}$ values in treated wells to the values observed in untreated controls. All dose-response assays were performed in technical triplicate and relative growth levels were averaged across replicates.

**Beneficial rhizobacteria dose-response assays.** P. simiae WSC417 and P. defensor WSC374r strains were provided by Keiko Yoshioka (University of Toronto). Saturated overnight cultures were diluted in LB to an $OD_{600}$ of 0.01 and 100 μL of culture was dispensed into each well of 96-well flat-bottom culture plates. Compounds were added to the wells using a multichannel pipette to a final DMSO concentration of 0.6% (v/v). The plates were sealed with clear plastic adhesive film and incubated at 28 °C with continuous shaking for 16 h. Endpoint growth was quantified be measuring the $OD_{600}$ with a microplate spectrophotometer (BioTek Epoch 2). The relative bacterial growth in each chemical exposure condition was calculated by dividing the endpoint $OD_{600}$ reading by the endpoint $OD_{600}$ reading in the DMSO solvent control. Three biological replicates were performed with three technical replicates on each day and the relative growth was averaged across the replicates.

**HEK293 cell dose-response assays.** HEK293 cells were seeded into 96-well plates, at 5000 cells per well, in 100 μL total volume of DMEM/10%FBS/1%PS media and grown overnight at 37 °C in the presence of 5% $CO_2$. Compounds were added to cells (final DMSO concentration of 0.5% v/v), and growth was continued for an additional 48 h. Following growth, 10 μL of CellTiter-Blue Viability reagent (Promega) was added to each well, and plates were incubated for 4 h at 37 °C in the presence of 5% $CO_2$. Fluorescence measurements (560 nm excitation/590 nm emission) were then performed using a CLARIOstar Plate Reader (BMG Labtech) to quantify reagent reduction and estimate cell viability. Fluorescence measurements were corrected for background from medium. Relative growth was calculated by dividing corrected fluorescence in the treatment wells by that measured in the corresponding DMSO control well. At least three replicates were performed and the relative growth was averaged across the replicates.

**HepG2 cell dose-response assays.** HepG2 cells (ATCC, Cat# HB-8065) were counted using a haemocytometer and diluted to 5 × 10⁴ cells/mL in 100 μL of RPMI-1640 (Sigma) medium supplemented with 10% heat inactivated fetal bovine serum (Gibco) following a standard protocol. Cells were seeded in black, clear-bottom 384-well plates (Corning) to a final density of 2000 cells/well in 40 μL. Cells were incubated at 37 °C with 5% $CO_2$ for 24 h. Subsequently, a 2-fold dilution series of test compound was added to seeded cells from 0 μM to 50 μM, and plates were incubated at 37 °C with 5% $CO_2$ for 72 h. After 72 h, Alamar Blue (Invitrogen) was added to the HepG2 cells at a final concentration of 0.05X and plates were incubated at 37 °C for 3 h. Fluorescence was measured at Ex560nm/Em590nm using a TECAN

Spark microplate fluorometer and values were corrected for background from the medium. Relative proliferation was calculated by dividing corrected fluorescence in the treatment wells by that measured in the corresponding DMSO control well. Three technical replicates were performed per experiment, and the relative proliferation was averaged across the replicates. The data presented is representative of two biological replicate experiments.

**Danio rerio (zebrafish) dose-response assays.** AB strain fish were maintained at 28.5 °C on a 14-10-h light–dark cycle and staged according to hours post fertilization (hpf). Eggs were collected at 4 hpf. At 3 days post fertilization (dpf), both male and female embryos were arrayed in 24-well culture plates, with 10 embryos per well. Next 5 μL of chemical dissolved in DMSO at the appropriate concentration or DMSO alone was added to 1 mL of E3 medium and vortexed to mix. Water was removed from the embryos in the wells and 1 mL of chemical-treated medium was transferred to each of the wells. The final DMSO concentration was 0.5% (v/v). The culture plates were sealed with Parafilm, wrapped in aluminum foil, and incubated at 28.5 °C for 72 h. At the assay endpoint, lethality was scored using a stereomicroscope either by appearance of whole-body necrosis or the absence of both a heartbeat and touch-evoked response. Relative viability was calculated by dividing the number of viable embryos in the treatment wells by the number of viable embryos in the DMSO control wells. Three biological replicates were performed, and the relative viability values were averaged across the three replicates. The zebrafish ethics protocol is 65697, approved by the Animal Care Committee at The Hospital for Sick Children, Canada. Fish were chosen at random for inclusion in the dose-response assays.

**Drosophila melanogaster dose-response assays.** Fly food in agar substrate was prepared by combining 50 mL of unsulfured molasses, 50 mL of cornmeal, 20.6 g of baker's yeast and 7.4 g of agar into 700 mL of distilled deionized water and boiling for 30 min. The media was cooled to 56 °C and 5 mL was added to cylindrical plastic fly vials. Next 10 μL of chemical dissolved in DMSO to the appropriate concentration, or DMSO alone, was added to the medium in each vial for a final DMSO concentration of 0.2% (v/v). The chemicals were mixed into the media using a pipette. The medium was allowed to solidify at room temperature overnight. The following day (day 0), 8 pairs of male and female w1118 flies were added to each vial. The vials were stored at room temperature for 6 days, at which point the number of motile flies was counted to assess adult fly viability. Fly motility was scored as any observable movement after the vial was vigorously tapped on the benchtop. Relative motility was calculated by dividing the average number of motile flies in the chemical treatment vials by the average number of motile flies in the DMSO control vials across three biological replicates. On day 6, the 16 parental flies were removed from the vials and the progeny larvae were allowed to continue to grow and hatch into adult flies. To assess fecundity and larval viability, hatched progeny flies were counted and discarded on days 13, 16, and 21 and the counts were summed. Relative larval viability was calculated by dividing the average number of hatched flies in the treatment vials by the average number of hatched flies in the DMSO control vials across three biological replicates.

#### POR knockdown dose-response experiments
*E. coli* HT115(DE3) containing an RNAi feeding vector expressing dsRNA targeting the *emb-8* gene or an empty RNAi feeding vector (L4440) from the Ahringer *C. elegans* RNAi feeding library distributed by Source BioScience Ltd. were grown overnight in LB with 100 μg/mL carbenicillin at 25 °C with no shaking until the culture was in mid-log phase (OD$_{600}$ ~ 0.6). The culture was induced with 1 mM IPTG and grown at 37 °C with shaking at 200 rpm for 1 h. The bacteria were pelleted and concentrated ten-fold with liquid NGM containing 1 mM

IPTG and 100 μg/mL carbenicillin. This bacterial suspension was dispensed into the wells of a flat-bottomed 96-well culture plate (40 μL per well). *C. elegans* temperature sensitive mutant strain MJ69 *emb-8(hc69)* and wild-type strain N2 synchronized L1s were obtained from an embryo preparation that was left to hatch at 15 °C overnight. Approximately 20 N2 L1s were plated in 10 μL of M9 per well containing L4440 empty vector control bacteria (wild-type condition). Approximately 20 *emb-8* L1s were plated in 10 μL of M9 per well containing *emb-8* dsRNA expressing bacteria (POR knockdown condition). Worms were grown for 40 h at the restrictive temperature of 25 °C with shaking at 200 rpm at which point 120 μM FUDR was added to each well and plates were returned to the 25 °C incubator for another 30 h. At this 72-h timepoint the compounds were added to the wells using a multichannel pipette to a final DMSO concentration of 1% v/v. After 48 h of chemical exposure viability was quantified, and the data is presented as the proportion of viable worms in each condition relative to the DMSO controls. Three biological replicates were completed with at least two technical replicates on each day, and the proportion viable was averaged across the replicates.

#### C. elegans P450 RNAi screen
The P450 RNAi screen consisted of RNAi-mediated knockdown of 69 *C. elegans* microsomal P450 genes. Each P450 RNAi feeding vector and the L4440 empty vector control from the Ahringer *C. elegans* RNAi feeding library distributed by Source BioScience Ltd. were grown in LB with 100 μg/mL carbenicillin at 25 °C with no shaking until the culture was in mid-log phase (OD$_{600}$ ~ 0.6). The culture was induced with 1 mM IPTG and grown at 37 °C with shaking at 200 rpm for 1 h. The bacteria were pelleted and concentrated ten-fold with liquid NGM containing 1 mM IPTG and 100 μg/mL carbenicillin. This bacterial suspension was dispensed into the wells of a flat-bottomed 96-well culture plate (40 μL per well). Approximately 20 N2 L1s obtained from an embryo preparation were plated in 10 μL of M9 per well. Worms were grown for 48 h at 20 °C with shaking at 200 rpm at which point 120 μM FUDR was added to each well and plates were returned to the 20 °C incubator for another 24 h. At this 72-h timepoint, 40 μM of cyprocide-B was added to each well (1% DMSO v/v). After 48 h of chemical exposure, the proportion of mobile worms was quantified in each condition. Four replicate wells were analyzed for each P450 knockdown condition, and the mobility was averaged across the replicates.

#### cyp-35D1 mutant dose-response experiments
*E. coli* HB101 was grown overnight in LB at 37 °C with shaking at 200 rpm. The saturated bacterial culture was pelleted and concentrated two-fold with liquid NGM. This bacterial suspension was dispensed into the wells of flat-bottomed 96-well culture plates (40 μL per well). *C. elegans* wild-type N2, *cyp-35D1(ean221)*, and *cyp-35D1(ean222)* synchronized L1s were obtained from an embryo preparation the previous day. Approximately 20 L1s were plated in 10 μL of M9 per well. Worms were grown for 48 h at 20 °C with shaking at 200 rpm at which point 120 μM FUDR was added to each well and plates were returned to the 20 °C incubator for another 24 h. At this 72-h timepoint, the compounds were added to the wells using a multichannel pipette to a final DMSO concentration of 1% v/v. After 48 h of chemical exposure, viability was quantified, and the data were presented as the proportion of viable worms in each condition relative to the DMSO controls. These dose-response experiments were completed in three biological replicates with two technical replicates on each day, and the relative viability was averaged across the replicates.

#### NACET experiments
**C. elegans NACET assays.** *C. elegans* L1+/− NACET viability assays were set up in liquid NGM in 96-well plate format as described above. NACET was dissolved directly in the + NACET media before dispensing at a final concentration of 5 mM and plates were incubated at 20 °C

with shaking at 200 rpm for 4 h. After 4 h, 25 μM of nematicide (cyprocide-B, wact-11 or wact-55) or DMSO solvent were added to the wells (1% final concentration of DMSO) and returned to the 20 °C incubator. After 24 h of drug exposure, the proportion of mobile (living) and immobile worms in each condition was quantified manually under a dissection microscope. Three biological replicates with at least three technical replicates of each condition were performed, and the proportion mobile worms relative to DMSO solvent controls are reported.

**D. dipsaci NACET assays.** *D. dipsaci* +/− NACET viability assays were set up in distilled water in 96-well plate format as described above. NACET was dissolved directly in the + NACET media before dispensing at a final concentration of 90 mM and plates were incubated at 20 °C with shaking at 200 rpm for 4 h. After 4 h, 60 μM of cyprocide-B, 30 μM of fluopyram or DMSO solvent were added to the wells (1% final concentration of DMSO) and returned to the 20 °C incubator. After 3 days of drug exposure, the proportion of mobile (living) and immobile worms in each condition was quantified manually under a dissection microscope after the addition of 2 μL of 1 M NaOH to the wells. Three biological replicates with at least three technical replicates of each condition were performed, and the proportion mobile worms relative to DMSO solvent controls are reported.

## 1-ABT experiments
1-ABT viability assays were set up in 100 μL liquid NGM + HB101 media (*C. elegans*) or 100 μL distilled water (PPN species) in 96-well plate format as described above. 1-ABT or DMSO solvent was added to a final concentration of 1 mM (0.5% DMSO v/v) in the media. Sealed plates were incubated at 20 °C with shaking at 200 rpm for 24 h. After 24 h 20 μM (*C. elegans*) or 60 μM (PPNs) final concentration of cyprocide-B or DMSO was added to the well (0.5% DMSO v/v). After an additional two (*M. incognita*) or three days (all other species) of compound exposure, the proportion of mobile (viable) worms in each condition was quantified manually under a dissection microscope. For *M. incognita*, two biological replicates were completed, each with six technical replicates. The proportion of mobile J2s relative to DMSO or 1-ABT alone controls is reported for each of the 12 replicates. For all other species, four biological replicates were performed with at least three technical replicates of each condition per day, and the average proportion mobile worms relative to DMSO or 1-ABT alone controls across the technical replicates is reported.

## Meloidogyne incognita soil-based root infection assays
Ninety grams of soil (1:1 sand:loam mix) was added to each compartment of 6-compartment plastic garden packs. The soil was drenched with 18 mL of deionized water containing dissolved chemical or DMSO solvent alone. 500 infective *M. incognita* second-stage juveniles (J2s) (strain IZ-1 originally collected from Veneta, Oregon; culture maintained and J2s collected as described in ref. 29) were then added to the soil in 2 mL of water, for a final chemical concentration of 60 μM in 20 mL of water (0.1% DMSO v/v). The J2s were incubated in the soil and chemical for 24 h, after which a 3-week-old tomato seedling was transplanted into each compartment. Plants were grown for 6 weeks in a greenhouse as described[29] under long-day conditions (16-h photoperiod) with 26/18 °C day/night temperatures. After 6 weeks, the plants were harvested and the roots gently washed with water. Eggs were extracted from the roots using a 10% bleach solution with agitation at 300 rpm for 3 min. The roots were rinsed over nested 250- and 25.4-μm sieves and eggs were collected in water from the fine sieve. The number of eggs found in the root system of each plant was counted using a haemocytometer under a dissection microscope. After egg extraction, the roots were dried in a 65 °C oven and the dry roots were

weighed. The number of eggs per milligram of root was calculated for each plant by dividing the number of eggs quantified by the mass of the dried root material. The infection assays were repeated on two different days, and three plants were analyzed from each chemical exposure condition on each day.

## Construction of P450-expressing yeast strains
The *C. elegans cyp-35D1* cDNA sequence was obtained from WormBase (WormBase web site, http://www.wormbase.org, release WS280, April 7, 2021) and codon-optimized for *S. cerevisiae* (see Supplementary Data 3 for sequence). The codon-optimized cDNA was synthesized by Integrated DNA Technologies. Using standard restriction cloning techniques, the *cyp-35D1* cDNA was cloned into the ATCC p416 GAL1 expression vector (URA3 selection marker; CEN6/ARSH4 origin of replication) and transformed into *S. cerevisiae* BY4741 (MATa his3Δ1 leu2Δ0 met15Δ0 ura3Δ0). Transformants were selected for on URA-SD media.

Nineteen P450s that were representative of the diversity of sequences within the *M. incognita* P450 superfamily were selected for analysis[8]. These 19 *M. incognita* P450 sequences were codon optimized for *S. cerevisiae* expression (see Supplementary Data 3 for sequences) and constructed by Twist Bioscience using the same ATCC p416 GAL1 expression vector. The 19 P450 expression vectors were individually transformed into *S. cerevisiae* BY4741 (MATa his3Δ1 leu2Δ0 met15Δ0 ura3Δ0), and transformants were selected for on URA- SD media.

## Dose-response experiments with P450-expressing yeast strains
Overnight cultures of *S. cerevisiae* strains containing either the *C. elegans* CYP-35D1 p416 GAL1 expression vector, the *M. incognita* CYP4731A3 p416 GAL1 expression vector or an empty vector control (strain construction described above) were set up in SD media with URA- selection containing 2% raffinose at 30 °C with shaking. The cultures were diluted to an $OD_{600}$ of 0.03 in SD URA- with 2% raffinose and grown for 4 h at 30 °C with shaking. To induce P450 expression, galactose was spiked into the culture at a 2% final concentration and grown for a further 4 h at 30 °C with shaking. 200 μL of culture was dispensed into each well of a 96-well flat-bottom culture plate. Compounds or DMSO solvent control were added to the wells using a multichannel pipette to a final DMSO concentration of 1% (v/v). The plates were sealed with clear plastic adhesive film and incubated at 30 °C with no shaking in a microplate spectrophotometer (BioTek Epoch 2) with $OD_{600}$ readings taken of each well every 30 min for 48 h. The area under the growth curve was calculated for each chemical exposure condition and divided by that of the DMSO solvent controls for the strain to produce a value for the relative growth in each condition. Four replicates of the dose-response assays were performed, and the relative growth values presented are the average across the replicates.

## Screen for M. incognita P450s that bioactivate cyprocide-B
Overnight cultures of the 19 *M. incognita* P450-expressing yeast strains (construction described above) and the empty vector (EV) control strain were set up in SD media with URA- selection containing 2% raffinose at 30 °C with shaking. The cultures were diluted to an $OD_{600}$ of 0.05 in SD URA- with 2% raffinose and grown for 2 h at 30 °C with shaking. To induce P450 expression, galactose was spiked into the culture at a 2% final concentration and grown for a further 2 h at 30 °C with shaking. 100 μL of culture was dispensed into each well of a 96-well flat-bottom culture plate and each strain was treated with 50 μM cyprocide-B or DMSO solvent control (1% v/v). The plates were sealed with clear plastic adhesive film and incubated at 30 °C with intermittent shaking in a microplate spectrophotometer (TECAN) with $OD_{600}$ readings taken of each well every 15 min for 48 h. Six replicates were completed for each condition. The growth data was background corrected by subtracting the $OD_{600}$ reading from the first timepoint

from all points in the growth curve. The timepoint at which the $OD_{600}$ was equal to 95% of the maximum of the curve was determined in the DMSO solvent control for each strain and the area under the curve (AUC) was calculated up to that timepoint for both the DMSO control and paired cyprocide-B-treated condition for the strain. The AUC for the cyprocide-B treatment condition divided by the AUC of the DMSO control is reported as the normalized AUC in response to cyprocide-B for each strain.

## Incubations for HPLC and LC-MS experiments

**C. elegans incubations for HPLC analysis.** Wild-type N2 and *emb-8(hc69)* L1s, and empty vector (L4440) and *emb-8* targeting RNAi bacteria were grown and prepared as described for the POR knockdown dose-response experiments. For the wild-type condition, 2 mL of L4440 bacterial suspension and 500 μL of M9 buffer containing 1250 N2 L1s were added to a 15 mL conical tube. For the POR knockdown condition, *emb-8* targeting RNAi bacteria and *emb-8(hc69)* L1s were used. Tubes were incubated on a nutator for 40 h at 25 °C at which point 120 μM FUDR was added to the tubes, and they were returned to 25 °C for another 30 h. At the 72-h timepoint, 100 μM cyprocide-B or DMSO were added to the tubes (0.5% DMSO v/v) and incubated for 6 h on a nutator at 25 °C. After the incubation, the worms were washed three times with M9 buffer and transferred to the wells of a Pall AcroPrep 96-well filter plate (0.45-μm wwPTFE membrane, 1 mL well volume). The wells were cleared by vacuum, worms were resuspended in 50 μL of M9 buffer, transferred to a 1.5 mL microcentrifuge tube, and immediately stored at −80 °C. Three replicates were performed for each condition.

**C. elegans and PPN incubations for LC-MS analysis of lysate.** Synchronized *C. elegans* wild-type N2 L1s were obtained using an embryo preparation. In 500 μL of M9 buffer, 150,000 L1s were incubated with 100 μM cyprocide-B or DMSO (0.5% v/v). Microcentrifuge tubes were incubated on a nutator for 6 h at 20 °C. After the incubation, the worms were transferred to the wells of a Pall AcroPrep 96-well filter plate (0.45-μm wwPTFE membrane, 1 mL well volume). The wells were cleared by vacuum and washed once with 600 μL of M9 buffer. The worms were resuspended in 50 μL of M9 buffer, transferred to a 1.5 mL microcentrifuge tube, and immediately stored at −80 °C. The *D. dipsaci* were collected as described above, and incubations were carried out similarly but with the following modifications: the incubations were performed in 2.5 mL of distilled water in 15 mL falcon tubes, 2500 *D. dipsaci* were used per incubation, and the incubation length was 24 h. *P. penetrans* were collected as described above, and incubations were carried out similarly to *D. dipsaci* but 20,000 mixed stage *P. penetrans* were used per incubation and the incubation length was 16 h. *M. hapla* J2s were collected as described above, and incubations were performed identically to *P. penetrans* but 15,000 *M. hapla* were used per incubation.

**D. dipsaci incubations for +/− 1-ABT LC-MS analysis.** *D. dipsaci* incubations for the +/− 1-ABT LC-MS analysis of incubation buffer were performed similarly to the incubations for lysate analysis with some modifications: the incubations were performed in a 1.5 mL microcentrifuge tube with 2000 worms in a 1 mL volume of distilled water. The worms were incubated with a 1 mM final concentration of 1-ABT or DMSO solvent (0.25% DMSO v/v) for 24 h before adding 100 μM cyprocide-B or paired DMSO solvent (0.5% DMSO v/v) and incubated for an additional 24 h. After the incubation, the worms were transferred to the wells of a Pall AcroPrep 96-well filter plate (0.45-μm wwPTFE membrane, 1 mL well volume), and the incubation buffer was cleared by vacuum into a collection plate. These incubation buffers were transferred to 1.5 mL microcentrifuge tubes and immediately frozen at −80 °C.

**Yeast incubations for lysate LC-MS analysis.** Overnight cultures of *S. cerevisiae* strains containing either the *C. elegans* CYP-35D1 p416 GAL1 expression vector, the *M. incognita* CYP4731A3 p416 GAL1 expression vector or an empty vector control (strain construction described above) were set up in SD media with URA- selection containing 2% raffinose at 30 °C with shaking. The cultures were diluted to an $OD_{600}$ of 0.3 in SD URA- with 2% raffinose in a 4.95 mL final volume in glass test tubes. The yeast cultures were grown for 4 h at 30 °C on a rotator. To induce P450 expression, 550 μL of 20% galactose was spiked into the culture for a 2% final concentration and grown for a further 4 h at 30 °C on a rotator. 100 μM cyprocide-B or cyprocide-E were added to the cultures (1% DMSO v/v). The cultures were vortexed and incubated for 6 h at 30 °C on a rotator. At the end point, the cells were transferred to 1.5 mL microcentrifuge tubes and pelleted, and all media was aspirated. The cells were washed in 0.5 mL of water, pelleted, and the water was removed. The cells were resuspended in 0.5 mL of M9 buffer and pelleted once again. All the media was removed, and the cell pellet was stored at −80 °C.

## Sample lysis for HPLC and LC-MS experiments

*C. elegans* worm pellets were thawed and 50 μL of 2X worm lysis buffer (20 mM Tris-HCl pH 8.3, 0.2% SDS, 240 μg/mL proteinase K) was added to each tube. Tubes were vortexed and incubated in a water bath at 56 °C for 80 min, vortexing every 20 min Tubes were bath sonicated for 20 min in a Branson 1510 bath sonicator at room temperature and lysates were frozen at −80 °C. PPN worm pellets were lysed using a similar methodology to *C. elegans* with the following modifications: the 2x worm lysis buffer was composed of 20 mM Tris-HCl pH 8.3, 0.2% SDS and 720 μg/mL proteinase K, and after lysis buffer was added the tubes were incubated in a water bath at 65 °C for 2 h (*D. dipsaci*) or 3 h (*M. hapla* and *P. penetrans*) with vortexing every 15 min.

The volume of the frozen *S. cerevisiae* pellets was adjusted to 50 μL with a solution of M9 buffer containing 1 M sorbitol and 300 U/mL zymolase. The cells were incubated at 37 °C for 60 min with vortexing every 10 min. 50 μL of 2X yeast lysis buffer (20 mM Tris-HCl pH 8.3, 0.2% SDS, 720 μg/mL proteinase K) was added to the tubes and the cells were incubated for 2 h at 56 °C with vortexing every 10 min. Tubes were bath sonicated for 20 min at room temperature and lysates were frozen at −80 °C.

## Incubation buffer processing for LC-MS analysis

*D. dipsaci* incubation buffers were processed for LC-MS analysis using Sep-Pak Light C8 Cartridges (Waters) fitted to a 10 mL BD Luer-Lok Syringe using a flow rate of -1 mL per minute. The column was activated with 3 mL of 100% acetonitrile (ACN) and then washed with 3 mL of Milli-Q water. The incubation buffer was then passed through the column and the column was washed with 1 mL of water. Three sequential elutions were performed using 250 μL each of 20%, 50% and 100% ACN. The eluates were combined, dried using an Eppendorf Vacufuge Concentrator set at 60 °C and stored at −80 °C.

## HPLC-DAD analysis and quantification

The frozen *C. elegans* wild-type and POR knockdown worm lysates were thawed and 50 μL of acidified acetonitrile (ACN) solution (50% ACN, 0.2% acetic acid) was added. The samples were vortexed and centrifuged at 17,949 × *g* for 1 min. Chromatographic separation was performed by reversed-phase chromatography (ZORBAX SB-C8 column, 4.6 × 150 mm, 5-micron particle size, with ZORBAX RX-C8 4.6 × 12.5 mm guard column) using an HP1050 system equipped with an autosampler, vacuum degasser, and variable wavelength diode-array detector (DAD). The column was maintained at -22 °C. 50 μL of lysate sample was injected and eluted over 8.65 min with the solvent and flow rate gradients shown in Supplementary Table 1. Absorbance was measured every 2 nm between 190 and 602 nm. Prior to processing the worm lysates, 20 nmol of cyprocide-B was processed to determine

elution time and absorbance spectrum. HPLC analyses of unmodified cyprocide-B parent and the five metabolites visible by HPLC-DAD (M1-M5) in worm lysates were performed three times, and area under the curve (AUC) for each analyte was quantified using the HP ChemStation peak integration tool, using default settings. To control for differences in biomass between samples all AUC values from a particular lysate were divided by the AUC of the endogenous worm contents in the lysate[21] and this value is reported as the normalized abundance. AUCs for the unmodified cyprocide-B parent and metabolite M3 were calculated at the absorbance intensity maximum of 280 nm, AUC for M1 and M2 were calculated at 300 nm, M4 at 250 nm, and M5 at 245 nm.

## LC-MS analysis and quantification

**Lysate LC-MS analysis.** Frozen *C. elegans* L1, *D. dipsaci, P. penetrans, M. hapla*, and yeast lysates were thawed and 50 μL of acidified acetonitrile (ACN) solution (50% ACN, 0.2% acetic acid) was added. The samples were vortexed and centrifuged at 17,949 × $g$ for 2 min. Chromatographic separation was performed by reversed-phase chromatography (Kinetex C18 analytical column, 100 × 2.1 mm, 2.5-Micron particle size, and ZORBAX C18 3.5-Micron particle size guard column) using an HPLC system (Agilent 1260 Infinity series binary pump with G1367D 1200 series HP autosampler) and mobile phase A (H₂O, 0.1% formic acid) and B (ACN, 0.1% formic acid). Five μL of lysate sample was injected and eluted with the solvent and flow rate gradients described in Supplementary Table 2. Electrospray ionization mass spectrometry (ESI-MS) analyses were carried out using an Agilent 6538 UHD quadrupole time-of-flight mass (Q-TOF) analyzer. The Q-TOF instrument was operated in positive scanning mode (90–2000 $m/z$) with the following parameters: VCap, 3500 V; fragmentor, 175 V; gas temperature, 325 °C; drying gas, 8 L/min; nebulizer, 30 psig. The Agilent MassHunter Qualitative Analysis software (version 10.0) was used for EIC extraction, peak integration and accurate mass measurement (Supplementary Table 3).

Tandem MS/MS was performed with the chromatographic separation and source parameters described above, using the targeted MS/MS acquisition mode of the instrument with isolation width set to Medium (~4 amu) and collision energy set to 20 or 30 eV. LC-MS/MS of nematode and yeast lysates was run by the AIMS Mass Spectrometry Laboratory in the Department of Chemistry at the University of Toronto, Canada.

**Buffer LC-MS analysis.** Dried buffer samples were solubilized in 120 μL of acidified acetonitrile (ACN) solution (20% ACN, 0.1% formic acid). The samples were vortexed and centrifuged at 17,949 × $g$ for 2 min. Chromatographic separation was performed by reversed-phase chromatography (ZORBAX Eclipse Plus C18 column, 2.1 × 50 mm, 1.8-Micron particle size, and 2.1 × 5 mm guard column) using an Agilent 1260 Infinity II HPLC system and mobile phase A (H₂O, 0.1% formic acid) and B (ACN, 0.1% formic acid). The column compartment was maintained at 40 °C. 2.5 μL of sample was injected at 100% A and 0.250 mL/min flow, followed by a linear gradient to 75% B over 22 min (0.250 mL/min flow), and finally a linear gradient to 100% B over 1 min (0.250 mL/min flow). ESI-MS analyses were carried out using an Agilent 6545 quadrupole time-of-flight mass analyzer operated in positive scanning mode (100–1000 $m/z$) with the following parameters: VCap, 3500 V; fragmentor, 110 V; gas temperature, 275 °C; drying gas, 10 L/min; nebulizer, 35 psig; sheath gas temperature, 350 °C; sheath gas flow, 12 L/min. The Agilent MassHunter Qualitative Analysis software (version 10.0) was used for EIC extraction, integration and quantification of EIC peak area.

## Reporting summary

Further information on research design is available in the Nature Portfolio Reporting Summary linked to this article.

## Data availability

All data are available in the main text, the Supplementary Information and/or in the Source data file. All data are available from the corresponding author upon request. Source data are provided with this paper.

## Code availability

The Python scripts for generating the heatmaps of HPLC chromatograms are published[41].

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

## Acknowledgements

The authors thank Benjamin Mimee (Agriculture and Agri-Food Canada) and Nathalie Dauphinais (Agriculture and Agri-Food Canada) for providing the *Meloidogyne hapla* and *Pratylenchus penetrans* cultures and for advice on culture methods; Qing Yu (Agriculture and Agri-Food Canada) for providing the *Ditylenchus dipsaci* culture and for advice on culture methods; Keiko Yoshioka (University of Toronto) for providing the *Pseudomonas simiae* and *Pseudomonas defensor* strains; Zhantao Shao and Henry Krause for help raising *D. melanogaster*; and Nicole Robbins for advice on the *S. cerevisiae* P450 expression assays. Some *C. elegans* strains were provided by the CGC, which is funded by NIH Office of Research Infrastructure Programs (P40 OD010440). We note that the USDA is an equal opportunity provider and employer. Funders include the Natural Sciences and Engineering Research Council of Canada Alexander Graham Bell Canada Graduate Scholarship (J.K.); EvoFunPath fellowship through the Natural Sciences and Engineering Research Council of Canada Research and Training Experience (CREATE) program 555337-2021 (E.P.); Canada Research Chair (Tier 1) grant in Microbial Genomics & Infectious Disease (L.E.C.); NIH R01 grant number AI153088 (E.C.A.); Canadian Institutes of Health Research Project Grants 480426 (J.J.D.); Canada Research Chair (Tier 1) grant in Chemical Biology (P.J.R.); Canadian Institutes for Health Research Foundation grant FDN-154288 (L.E.C.); Canadian Institutes of Health Research Project Grants 173448 and 186156 (P.J.R.).

## Author contributions

Conceptualization: J.K., A.R.B., and P.J.R. Methodology: J.K., A.R.B., and P.J.R. Investigation: J.K., A.R.B., B.C., S.R.C., M.K., J.C., J.M.P.C., E.P., J.S., E.K., J.B.C., and S.G. Visualization: J.K., A.R.B., and P.J.R. Funding acquisition: P.J.R., I.Z., M.L., L.E.C., I.S., E.C.A., and J.J.D. Project administration: P.J.R. Supervision: P.J.R., I.Z., M.L., L.E.C., I.S., E.C.A., and J.J.D. Writing—original draft: J.K. Writing—review & editing: J.K. and P.J.R.

## Competing interests

L.E.C. is co-founder and shareholder in Bright Angel Therapeutics, a platform company for development of novel antifungal therapeutics, and is a Science Advisor for Kapoose Creek, a company that harnesses the therapeutic potential of fungi. Mention of trade names or commercial products in this publication is solely for the purpose of providing specific information and does not imply recommendation or endorsement by the U.S. Department of Agriculture. USDA is an equal opportunity provider and employer. J.K., A.R.B., B.C., and P.J.R. have a patent application related to the Cyprocides. The remaining authors declare no competing interests.
