## [Peer Review File · Nature Communications]

Cyprocide Selectively Kills Nematodes via Cytochrome P450 BioactivationREVIEWER COMMENTS

Reviewer #1 (Remarks to the Author):

The study entitled "Diverse Plant-Parasitic Nematodes are Selectively Killed by Oxadiazole Thioether Pro-Nematicides" is a continuation of the study "Selective control of parasitic nematodes using bioactivated nematicides" published in Nature by the same research group. However, this study identified and focused on structurally similar compounds with the same scaffold named disubstituted oxadiazole (DODA) core effectiveness via P450 enzyme activation. The P450s CYP35D1 and CYP4731A3 were found to be responsible for activating the compounds. One of these P450s, CYP4731A1, is also the prime candidate in their study previously published in Nature.

The study's results are novel and immensely contribute to understanding how the P450 enzymes can control nematodes. The methodology is detailed, and the results and discussion are concise; however, detailed information provided in the supplementary section variants the reproducibility of the results.

I strongly recommend the article for acceptance in the current format

Reviewer #2 (Remarks to the Author):

The manuscript describes the identification of the chemical Cyprocide which is bioactivated into a lethal reactive electrophilic metabolite by a specific nematode cytochrome P450 enzyme. The CYP450 of *M. incognita* responsible for activation has been identified. In this regard the manuscript follows an extremely similar pathway to that of the applicants' preceding Nature publication which they cite (Burns et al., 2023), with another chemical – in this manuscript Cyprocide, a multiple disubstituted oxadiazole.

The methodology and results are sound.

The claim "Cyprocide that selectively kills diverse plant-parasitic nematodes" is misleading and not accurate. There are many allusions to the specificity of the compound throughout the manuscript without concomitant results to validate them. *C. elegans* has been used in the methodology and is therefore known to be affected by Cyprocide. This ignores the important role that free living nematodes play in belowground food-webs. There is a stark

of lack of testing against other invertebrate species.

The discussion is not a serious attempt at describing the importance and placing the implications of the research findings in context with the literature. It is merely three short, scant and superficial paragraphs. This presumably reflects the fact that most 'Discussion' has already taken place in the previous manuscript.

Reviewer #3 (Remarks to the Author):

In their manuscript titled "Diverse Plant-Parasitic Nematodes are Selectively Killed by Oxadiazole Thioether Pro-Nematicides", Peter J. Roy et al. prospectively discovered that the 1,3,4-oxadiazole thioether compound Cyprocide could be metabolized into their electrophilic products by selective bioactivating nematode's P450s. To do this, the authors conducted a long-term study of nematicides discovery through triggering the cytochrome P450-mediated bioactivation. The authors found that multiple disubstituted oxadiazoles had excellent nematode-selective activity and thus assembled an expanded oxadiazoles' library to assess their nematocidal activity towards various nematodes. Notably, the authors indicated the action mechanism of cyprocide-B that selectively killed the nematodes, and these broad-acting P450-dependent 1,3,4-oxadiazole thioether were nontoxic activity towards non-target organisms. Overall, the authors use well-established tools and methods to answer their questions, and this manuscript may attract broad interests from multidisciplinary research communities for different applications. I support the publication of this study pending some concerned points.

MAJOR FINDINGS

The most interesting significant finding from this manuscript is how to discover these P450-dependented 1,3,4-oxadiazole thioether compounds (especially Cyprocide) and reveal their action mechanism. Secondly, the authors also prove that P450 bioactivation is a viable tactic for the selective killing of nematodes. Furthermore, the authors also make a notable contribution by activating the nematode's P450s to control nematode infection. Finally, the authors report that Cyprocides have the potential to meet this need and help secure global food production. Overall, the authors make a significant contribution to the new agrochemical discovery.

MAJOR CONCERNS

1. Nematodes employ water, soil, and insects as media to parasitize and reproduce. Therefore, whether cyprocide-B is also safe for fish or not?
2. To validate the hypothesis that cyprocide-B sulfoxide metabolite induces lethality by depletion of anti-oxidant LMW thiols (glutathione, γ -glutamylcysteine, cysteinylglycine, and cysteine), why did the author opt the exogenously molecule NACET rather than the LMW thiols directly?
3. Line 200, the authors claimed that the trifluoromethylated cyprocide-I analog exhibited eight-fold greater potency in vitro against *M. hapla* than cyprocide-B. Why not select it as a model drug instead of cyprocide-B?
4. In the "Supplementary Information" part, the figures of NMR, IR, and HRMS of target compounds are recommended to supplement. In addition, some recent literature about anti-Xoo activity may provide more information for the authors. (Advanced Functional Materials, 2023: 2303206; Chemical Engineering Journal, 2023, 464: 142432.)
5. As a nematicide that inhibits succinate dehydrogenase, why choose fluopyram as a positive control?

MINOR CONCERNS

1. The "plant-parasitic nematodes (PPNs)" uses the abbreviation in the section of Cyprocide Controls Root Infestation by *M. incognita*, but the full name in DISCUSSION section. It is, therefore, suggested that the author write the full name when using it for the first time and use abbreviations for the rest to improve readability. Besides, similar problems in the manuscript should be carefully checked and revised by the author.
2. In Figure 4, the layout of pictures in the manuscript is confusing.
3. In the references section, there are some formatting issues. For example, in Line 276, "Nat Commun." should be "Nat. Commun."; Line "350", "J Am Chem Soc." should be "J. Am. Chem. Soc.". Please check it carefully.
4. On page 6, replace "Fig 1A, Fig. 2I" with "Fig. 1A, Fig. 2I"; Similarly, "Fig 3D" should be changed to "Fig. 3D".

Reviewer 1-Opening Summary: The study entitled “Diverse Plant-Parasitic Nematodes are Selectively Killed by Oxadiazole Thioether Pro-Nematicides” is a continuation of the study “Selective control of parasitic nematodes using bioactivated nematicides” published in Nature by the same research group. However, this study identified and focused on structurally similar compounds with the same scaffold named disubstituted oxadiazole (DODA) core effectiveness via P450 enzyme activation. The P450s CYP35D1 and CYP4731A3 were found to be responsible for activating the compounds. One of these P450s, CYP4731A1 [*sic*], is also the prime candidate in their study previously published in Nature.

Reviewer 1-Comment 1a: The study's results are novel and immensely contribute to understanding how the P450 enzymes can control nematodes. The methodology is detailed, and the results and discussion are concise;

Our response: We thank the reviewer for the kind words.

Reviewer 1-Comment 1b: however, detailed information provided in the supplementary section variants the reproducibility of the results.

Our response: We apologize in advance, but we are not certain what variability the reviewer is referring to. Our best guess is that within the supplementary materials, Figure S3 shows some differences in the abundance of the cyproicide metabolites between the nematode species examined.

In particular, the sulfoxide metabolite was not detected in *M. hapla* or *D. dipsaci*, and the abundance of the cyproicide-B parent and LMW thiol conjugates varied between species.

These observations could be the result of differences in the cyproicide-B incubation times used with the different species because of their different responses to the compound (as outlined in the Figure S3 caption and the methods section).

These observations could also be the result of differences in available LMW thiol abundance between species, differences in drug bioavailability and efflux capability between species, differences in rate of metabolism and/or reactive metabolite conjugation between species, etc.

Due to the observed variability of metabolite abundance between the different species, we do not make any specific claims about metabolite abundance between the nematode species examined. Instead, in the relevant results section (~line 172 in the revised manuscript), we simply state that the masses detected by LCMS are consistent with cyproicide-LMW thiol conjugates being formed in each of the species examined, which suggests that cyproicide-B is bioactivated into a reactive electrophile in the PPNs similar to what is observed with *C. elegans*.

Reviewer 1-Comment 2: I strongly recommend the article for acceptance in the current format

Our response: We thank the reviewer for the encouragement.

Reviewer 2-Opening Summary: The manuscript describes the identification of the chemical Cyprocide which is bioactivated into a lethal reactive electrophilic metabolite by a specific nematode cytochrome P450 enzyme. The CYP450 of *M. incognita* responsible for activation has been identified. In this regard the manuscript follows an extremely similar pathway to that of the applicants' preceding Nature publication which they cite (Burns et al., 2023), with another chemical – in this manuscript Cyprocide, a multiple disubstituted oxadiazole.

Reviewer 2-Comment 1: The methodology and results are sound.

Our response: Thank you.

Reviewer 2-Comment 2: The claim "Cyprocide that selectively kills diverse plant-parasitic nematodes" is misleading and not accurate. There are many allusions to the specificity of the compound throughout the manuscript without concomitant results to validate them. *C. elegans* has been used in the methodology and is therefore known to be affected by Cyprocide. This ignores the important role that free living nematodes play in belowground food-webs.

Our response: We do not disagree with the reviewer's main point and apologize for our oversight-our intention was certainly not to mislead. We have addressed the issue in the following ways.

First, we have changed the title from, '*Diverse Plant-Parasitic Nematodes are Selectively Killed by Oxadiazole Thioether Pro-Nematicides*', to, '*Cyprocide Selectively Kills Nematodes via Cytochrome P450 Bioactivation*'.

Second, we examined all relevant sentences throughout and changed them where necessary. For example, in the abstract, we changed, '*Here, we report our discovery of a 1,3,4-oxadiazole thioether scaffold called Cyprocide that selectively kills diverse plant-parasitic nematodes.*', to, '*Here, we report our discovery of a 1,3,4-oxadiazole thioether scaffold called Cyprocide that selectively kills nematodes including diverse species of plant-parasitic nematodes.*'

Third, we have added a new paragraph in the discussion that addresses the reviewer's second point in the comment above:

The Cyprocides exhibit similar or improved selectivity for nematodes over many of the commercial nematicides previously tested in these non-target organism assays (Burns et al., 2023). However, as evidenced throughout this study, many of the Cyprocides examined are lethal to the free-living nematode C. elegans and thus may potentially pose a threat to soil-beneficial nematode communities. While the same can be said of other next-generation commercial nematicides including thioxazafen and fluopyram (this work, Burns et al., 2023, Burns et al., 2015), future efforts can be taken to identify Cyprocide analogs that minimize detrimental effects on free-living nematodes while maintaining activity in PPNs.

Finally, we assume the reader understands that we are not reporting on a product, but on a novel small molecule scaffold that requires further development if it were to be commercialized. We help reinforce that point in the final revised words of the discussion as follows: '*The discovery of the Cyprocides has reinforced the idea that P450 bioactivation is a viable mechanism through which this need may be met. Further development of the scaffold may ultimately help to secure global food production.*'

Reviewer 2-Comment 3: There is a stark of lack of testing against other invertebrate species.

Our response: In the original submission, we tested cyprocide-B against six different non-nematode systems, including two human cell lines, two fungal species, and two plant beneficial rhizobacteria. To address the reviewer's comment and increase the diversity of our non-target species testing we have since tested cyprocide-B's activity in adult and larval stages of the invertebrate insect species *Drosophila melanogaster* and in the zebrafish *Danio rerio*. These results are included in the revised

Figure 1E and described in the corresponding results section 'Cyprocide Selectively Kills Nematodes'.

Of note, we have also updated the language, especially in the discussion, to avoid any impression that we think that Cyprocide is ready to use in the field. Like many papers before ours, this manuscript describes a first-in class molecule with potential utility, and not a market-ready product, which is beyond the scope of our work.

Reviewer 2-Comment 4: The discussion is not a serious attempt at describing the importance and placing the implications of the research findings in context with the literature. It is merely three short, scant and superficial paragraphs. This presumably reflects the fact that most 'Discussion' has already taken place in the previous manuscript.

Our response: We apologize for the short discussion. It reflects the word limits that we aspired to in our initial submission to another journal, which was ultimately passed on to *Nature Communications* for review. The updated manuscript now has an expanded discussion.

Reviewer 3-Opening Summary: In their manuscript titled “Diverse Plant-Parasitic Nematodes are Selectively Killed by Oxadiazole Thioether Pro-Nematicides”, Peter J. Roy et al. prospectively discovered that the 1,3,4-oxadiazole thioether compound Cyprocide could be metabolized into their electrophilic products by selective bioactivating nematode’s P450s. To do this, the authors conducted a long-term study of nematicides discovery through triggering the cytochrome P450-mediated bioactivation. The authors found that multiple disubstituted oxadiazoles had excellent nematode-selective activity and thus assembled an expanded oxadiazoles’ library to assess their nematocidal activity towards various nematodes. Notably, the authors indicated the action mechanism of cyprocide-B that selectively killed the nematodes, and these broad-acting P450-dependent 1,3,4-oxadiazole thioether were nontoxic activity towards non-target organisms.

Reviewer 3-Comment 1: Overall, the authors use well-established tools and methods to answer their questions, and this manuscript may attract broad interests from multidisciplinary research communities for different applications. I support the publication of this study pending some concerned points.

Our response: We thank the reviewer for the encouragement.

Reviewer 3- MAJOR FINDINGS Summary: The most interesting significant finding from this manuscript is how to discover these P450-dependented 1,3,4-oxadiazole thioether compounds (especially Cyprocide) and reveal their action mechanism. Secondly, the authors also prove that P450 bioactivation is a viable tactic for the selective killing of nematodes. Furthermore, the authors also make a notable contribution by activating the nematode’s P450s to control nematode infection. Finally, the authors report that Cyprocides have the potential to meet this need and help secure global food production.

Reviewer 3-Comment 2: Overall, the authors make a significant contribution to the new agrochemical discovery.

Our response: These are very nice words- thank you!

Reviewer 3- Comment 3: Nematodes employ water, soil, and insects as media to parasitize and reproduce. Therefore, whether cyprocide-B is also safe for fish or not?

Our response: To address this comment, we have performed additional testing to assess cyprocide-B activity in 3-day post-fertilization zebrafish *Danio rerio*. We found that cyprocide-B only impacted fish viability at the highest concentration tested (50 μ M), and it outperformed the commercial nematicide tioxazafen in terms of fish safety in this dose-response analysis. These results are reflected in the updated Figure 1E and described in the corresponding results section ‘Cyprocide Selectively Kills Nematodes’.

We have made further efforts to increase the diversity of our non-target species testing by assaying cyprocide-B activity in adult and larval stages of the insect *Drosophila melanogaster*. These results are also reflected in the updated Figure 1E and described in the corresponding results section ‘Cyprocide Selectively Kills Nematodes’.

Reviewer 3- Comment 4: To validate the hypothesis that cyprocide-B sulfoxide metabolite induces lethality by depletion of anti-oxidant LMW thiols (glutathione, γ -glutamylcysteine, cysteinylglycine, and cysteine), why did the author opt to use the exogenously molecule NACET rather than the LMW thiols directly?

Our response: LMW thiols such as cysteine and glutathione have poor bioavailability and are unlikely to reach sufficient concentrations in cells. By contrast, N-acetylcysteine ethyl ester (NACET) is readily bioavailable, and as explained in the manuscript, is readily converted by the cell into cysteine, which is the rate limiting substrate for γ -glutamylcysteine and glutathione synthesis. Hence,

NACET is a single bioavailable reagent that readily increases the cellular stores of cysteine, γ -glutamylcysteine and glutathione (see Giustarini et al., 2012, which we refer to in the manuscript).

Reviewer 3- Comment 5: Line 200, the authors claimed that the trifluoromethylated cyproicide-I analog exhibited eight-fold greater potency *in vitro* against *M. hapla* than cyproicide-B. Why not select it as a model drug instead of cyproicide-B?

Our response: The simple answer is that we got the trifluoromethylated cyproicide-I results at the end of the study after several years of analyses with cyproicide-B.

Here is the longer answer: Cyproicide-B was the most potent nematicidal analog *in vitro* of the original set of cyproicide analogs in the DODA screening library. Furthermore, cyproicide-B exhibited a good selectivity profile among nematodes and non-target organisms. Thus, cyproicide-B was selected as a model analog to characterize the mode of action of the cyproicide scaffold and to decipher the CYP-catalyzed metabolism of the scaffold which is described in this manuscript.

It was only with these findings in hand that we were able to design and synthesize the expanded set of cyproicide analogs described in Figure S6 (including cyproicide-I) that would maintain the mode of action of the scaffold. These analogs were designed based on our hypothesis that they would increase the reactivity/electrophilicity of the CYP-modified sulfoxide metabolite and/or improve nematode bioavailability— and thus exhibit increased potency in the worm.

Now that we have identified cyproicide-I and several other cyproicide analogs with promising activity in *Meloidogyne*, we are following up with soil-based *Meloidogyne* tomato infection assays and additional activity profiling to examine the selectivity of these more potent analogs. This work is ongoing and may be the focus of a future manuscript.

Reviewer 3- Comment 6a: In the “Supplementary Information” part, the figures of NMR, IR, and HRMS of target compounds are recommended to supplement.

Our response: Figures showing NMR spectra have been added to the Supplementary Information for synthesized compounds 3a-3p. Figures showing HRMS spectra have been added to the Supplementary Information for those compounds that are new in the paper (3a, 3c-3l, 3n-3p). Figures showing IR spectra have not been included here as these spectra are typically not reported according to ACS journal guidelines. Numerical summaries are provided in the Supplementary Information for those compounds that are new in the paper (3a, 3c-3l, 3n-3p).

Reviewer 3- Comment 6b: In addition, some recent literature about anti-Xoo activity may provide more information for the authors. (Advanced Functional Materials, 2023: 2303206; Chemical Engineering Journal, 2023, 464: 142432.)

Our response: We thank the reviewer for the suggestion. We have investigated the cited papers and found that they focus on innovations in delivery systems that could be used for pesticides, as opposed to relevant pesticide scaffolds. As we want to keep our discussion focused on the Cyproicide scaffold, we felt it not germane to our story. We have also investigated previous publications that have shown anti-Xoo activity of disubstituted 1,3,4-oxadiazole scaffolds and found reports that it is the sulfone derivative of the scaffold that exhibits potent activity over the thioether. We don't think that these structures are relevant because the oxidized version of our scaffold (i.e., the sulfoxide) should be contained within the nematode and would not impact other pathogens in the soil.

Reviewer 3- Comment 7: As a nematicide that inhibits succinate dehydrogenase, why choose fluopyram as a positive control?

Our response: Fluopyram was used as a control in two different experiments in this manuscript.

First, Fluopyram was used **as a negative control** in our test of whether NACET could suppress cyprocide's lethal effects against *Ditylenchus dipsaci* (Figure 4E). We clarify its use in the revision by writing (new sentence underlined):

...we found that preincubation of D. dipsaci with NACET significantly suppressed the lethality induced by cyprocide-B ($p < 1E-6$) but not that conferred by the fluopyram nematicide control (Fig. 4E). Here, fluopyram was selected as a negative control because fluopyram kills D. dipsaci and inhibits its target without the need for bioactivation (Burns et al., 2015).

Second, fluopyram was used as a control in the soil-based tomato root infestation assays with *M. incognita* (described in the 'Cyprocide Controls Root Infestation by *M. incognita*' results section line ~201 in the revised manuscript and Figure 4J-K). Here, fluopyram was chosen **as a positive control** for PPN killing because of its previous effectiveness when applied as a soil drench in similar soil-based infestation assays (Burns, et al., 2023). The specific mode of action of fluopyram was not taken into consideration. We have updated the relevant text as follows (additions are underlined):

'The Cyprocides were tested alongside two next-generation soil-applied commercial nematicide controls, fluopyram and tioxazafen, which have proven to be effective at controlling M. incognita infestation in similar assays (Burns et al., 2023).

Reviewer 3- Comment 8: The "plant-parasitic nematodes (PPNs)" uses the abbreviation in the section of Cyprocide Controls Root Infestation by *M. incognita*, but the full name in DISCUSSION section. It is, therefore, suggested that the author write the full name when using it for the first time and use abbreviations for the rest to improve readability. Besides, similar problems in the manuscript should be carefully checked and revised by the author.

Our response: Thank you for pointing this out. This change has been made for PPNs as well as other abbreviations within the body of the manuscript. Note that we considered the abstract to be distinct, so we spelled out "plant-parasitic nematodes" in both the abstract and at first mention within the introduction, followed by PPNs for all other uses.

Reviewer 3- Comment 9: In Figure 4, the layout of pictures in the manuscript is confusing.

Our response: We have rearranged the panels in a more spatially logical manner. We hope that the changes are satisfactory.

Reviewer 3- MINOR CONCERN 3: In the references section, there are some formatting issues. For example, in Line 276, "Nat Commun." should be "Nat. Commun."; Line "350", "J Am Chem Soc." should be "J. Am. Chem. Soc.". Please check it carefully.

Our response: We have gone through the citations in detail and have corrected the errors.

Reviewer 3- MINOR CONCERN 4: On page 6, replace "Fig 1A, Fig. 2I" with "Fig. 1A, Fig. 2I"; Similarly, "Fig 3D" should be changed to "Fig. 3D".

Our response: These changes have been made.

REVIEWERS' COMMENTS

Reviewer #3 (Remarks to the Author):

The authors have improved the quality of paper and answer all the questions. I suggest the paper is ready for publication.

Editorial Note: [Please note that Reviewer #2 also provided their report (only comments to the editor which are not visible to the authors) but still does not consider the advance over your previous publication sufficient. We have editorially overruled this.]